# ENHANCING FOUNDATION MODELS FOR TIME SERIES FORECASTING VIA WAVELET-BASED TOKENIZATION

## ABSTRACT

There is a major open question about how to best develop foundation models for time series forecasting. *Tokenization* is a crucial consideration in this effort: what is an effective discrete vocabulary for a real-valued sequential input? To address this question, we develop WaveToken, a wavelet-based tokenizer that allows models to learn complex representations directly in the space of time-localized frequencies. Our method first scales and decomposes the input time series, then thresholds and quantizes the wavelet coefficients, and finally pre-trains an autoregressive model to forecast coefficients for the horizon window. By decomposing coarse and fine structures in the inputs, wavelets provide an eloquent and compact language for time series forecasting that simplifies learning. Empirical results on a comprehensive benchmark, including 42 datasets for both in-domain and zero-shot settings, show that WaveToken: *i)* provides better accuracy than recently proposed foundation models for forecasting while using a much smaller vocabulary (1024 tokens), and performs on par or better than modern deep learning models *trained specifically on each dataset*; and *ii)* exhibits superior generalization capabilities, achieving the *best average rank across all datasets* for three complementary metrics. In addition, we show that our method can easily capture complex temporal patterns of practical relevance that are challenging for other recent pre-trained models, including trends, sparse spikes, and non-stationary time series with varying frequencies evolving over time.

## 1 INTRODUCTION

Time series forecasting is integral to decision-making processes in many domains, including finance, healthcare, supply chain optimization, and climate science. Over the last decade, the field has seen a gradual but steady adoption of "global" deep learning models in lieu of traditional "local" statistical models (Lara-Benítez et al., 2021; Benidis et al., 2022). Recently, the success of large language models (LLMs) on natural language and vision applications has spurred an increasing interest for developing similar "foundation models" in other fields (for example, Subramanian et al. (2024); Golling et al. (2024); Ansari et al. (2024); Das et al. (2023)). These efforts aim at building general-purpose machines able to learn complex representations from vast amounts of data and to generalize to a wide variety of tasks, based on the premise that LLMs are general pattern recognizers (Mirchandani et al., 2023). In other words, if a problem can be reduced to that of modeling an arbitrary sequence of tokens defined on a discrete vocabulary, then an autoregressive transformer might be capable of learning non-trivial relationships via next-token prediction, regardless of the inputs representing text or not. The sequential nature of time series forecasting aligns seamlessly with this perspective, which is why several recent works have proposed adapting transformer-based architectures into foundation models for time series (see Section 2 for a review).

Tokenization is a crucial albeit still understudied (Dagan et al., 2024) component of LLMs, as it provides the vocabulary on which token streams are defined and the autoregressive structure is learned. While in principle transformers can learn arbitrary dependencies, in practice it matters whether the architecture can efficiently leverage specific structures in the inputs. In the context of time series forecasting, it is then important to answer the following question: what is the most appropriate *discrete* vocabulary for a *continuous* (real-valued) sequential input? In other words, what is the correct "language" for time series forecasting? As the goal is to develop a unified model with excellent forecasting performance on unseen datasets, an ideal dictionary of tokens should be as expressive as possible while also being compact, in order to efficiently represent the extremely high variety of non-

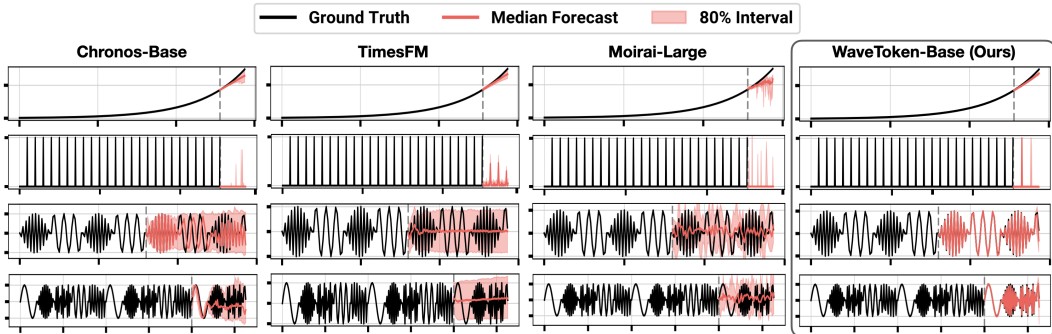

Figure 1: **`WaveToken-Base` (199M parameters) provides excellent forecasts with very low uncertainty.** Performance of different foundation models for time series forecasting on complex patterns of practical relevance: Chronos-Base (201M), TimesFM (200M) and Moirai-Large (311M) struggle to capture exponential trends (*top row*), sparse spikes (*second row*), and non-stationary signals with 2 and 5 frequencies evolving over time (*bottom two rows*).

stationary real-world time series. In addition, different "languages" can exhibit various dependency structures: although time series have a natural "order" dictated by time, one might ask whether there exists a mapping providing a more eloquent re-ordering in a suitable space that exposes important features and simplifies learning. Popular existing frameworks leverage temporal patches (e.g., Das et al. (2023)) or quantization into prescribed bins (e.g., (Ansari et al., 2024)) and lead to competitive performance in standard settings. However, by preserving the natural temporal dependency in the input, these approaches tend to focus more either on the recent history or uniformly over all time steps. As a result, they struggle to simultaneously capture both local and global patterns that often represent relevant cases in practical applications (see, e.g., Figure 1).

**Main contributions.** In this work, we propose and analyze a tokenizer based on wavelets, which are families of basis functions used to decompose a signal into small waves with high resolution in both time and frequency domain (Mallat, 2009). In particular, we develop a tailored pre- and post-processing tokenization pipeline — WaveToken — that allows the model to learn directly in the space of sparse wavelet coefficients. This yields a compressed but highly expressive vocabulary that facilitates the encoding of complex non-stationary time series. In addition, it provides an explicit multi-scale structure in the inputs, which the model learns to exploit by effectively forecasting from coarser to finer resolutions. We pair our wavelet-based tokenizer with the popular T5 encoder-decoder architecture of Raffel et al. (2020) and pre-train it on a large corpus of time series from different domains. We evaluate WaveToken on a comprehensive in-domain and zero-shot benchmark made of 42 real-world datasets and compare it with traditional statistical models, recent LLM-based forecasters, and state-of-the-art deep learning models. Empirical results show that

*i)* in terms of **accuracy**, WaveToken achieves superior forecasting performance and always performs on-par or better than all other baselines with respect to three complementary metrics, while using a *much smaller vocabulary* (1024 tokens) than recent foundation models for time series.

*ii)* WaveToken exhibits superior **generalization** capabilities, achieving the *best average rank across all datasets* for all metrics. In addition, our method can easily capture complex temporal patterns in several edge cases relevant for practical applications, such as exponential trends, sparse spikes and signals with different frequencies varying over time, as shown in Figure 1.

Overall, our findings not only suggest that wavelet-based tokenization can provide a compelling language for time series forecasting with LLMs, but also position our approach as a promising avenue for developing general-purpose, information-efficient forecasting models.

## 2 RELATED WORK

**Tokenization in LLMs.** Most tokenizers originate from the natural language processing (NLP) literature and have been developed and studied for various domains such as math (Singh & Strouse,

2024), code (Zheng et al., 2023), and several languages (e.g., Tolmachev et al. (2018); Alyafeai et al. (2023)). In modern LLMs, the most popular tokenizers learn a dictionary directly on data, such as variants of Byte-Pair Encoding (BPE; Gage (1994)). Tokenization of real-valued numbers has received particular attention due to the difficulty of finding what is the most appropriate discrete vocabulary for a continuous input. One of the most popular methods is training a Vector Quantized-Variational AutoEncoder (VQ-VAE) (Van Den Oord et al., 2017), which learns a dictionary of $k$-dimensional codewords to capture latent representations. Instead of learning a token for each numerical value, Golkar et al. (2023) proposes a fixed scheme that allocates a dedicated embedding vector for numerics and scales it by the number value, thereby improving efficiency and generalization. In this work, we similarly adopt a fixed scheme, but employ a tailored wavelet decomposition that enhances the crucial spectral properties of time series signals.

**Wavelet-based forecasters.** Several approaches have tried to embed wavelets in forecasting pipelines. An early example is that of Papadimitriou et al. (2003), who integrate the discrete wavelet transform with ARIMA modeling to capture complex patterns over long time periods. More recently, Zhou et al. (2022) proposed to substitute attention blocks with Fourier- and wavelet-enhanced blocks in the transformer architecture. Within this line of work, Zhang et al. (2022) proposed to model trend and seasonality separately with an MLP and Fourier attention, respectively. In addition, they showed that under *linear* transformations, attention models in time domain, Fourier domain and wavelet domain have the same representation power. Sasal et al. (2022) leverage a redundant wavelet transform that yields $J$ series of coefficients (one per decomposition level), each having the same length as the original time series. They then learn a separate transformer model for each scale. To the best of our knowledge, WaveToken is the first application of a maximally decimated wavelet transform to build a tokenizer tailored for time series forecasting with LLMs.

**Modeling and generation of signals.** Different methods in the signal processing literature propose to integrate spectral or wavelet decomposition with deep learning architectures to enhance the capabilities of these models on a variety of tasks. Apart from the already-mentioned VQ-VAE (Van Den Oord et al., 2017), several works exploit spectrograms and log-mel spectra, for example, as pre-processing steps on medical or audio data (Choi et al., 2023; Purwins et al., 2019). Similarly, audio codecs (Zeghidour et al., 2021) are emerging as a critical technique to bridge the gap between continuous waveforms and token-based language models. As for image generation, Guth et al. (2022) proposed to speed up denoising by learning diffusion models in the wavelets domain. Recent concurrent works by Tian et al. (2024), Mattar et al. (2024) and Zhu & Soricut (2024), learn an autoregressive model on a multi-scale sequence or in the space of 2D wavelet coefficients.

**Foundation models for time series forecasting.** Several approaches are being developed to adapt LLMs to other domains, such as time series forecasting. Recent efforts in this direction include, e.g., Xue & Salim (2023), which converts time series into text and re-frames the task as a question-answering problem; Gruver et al. (2024), which tokenizes real-valued data as strings of digits and leverages models such as GPT-3 (Brown, 2020) and Llama 2 (Touvron et al., 2023); and Jin et al. (2023), which prompts a frozen LLM with a prefix describing the task and patch embeddings of time series aligned with text prototypes. Recently, Rasul et al. (2023); Goswami et al. (2024); Das et al. (2023); Ansari et al. (2024); Woo et al. (2024); Talukder et al. (2024) proposed different paradigms to pre-train transformer-based architectures on a large corpus of time series. See also Zhang et al. (2024) for a recent survey. While these works adopt domain-specific designs such as patching, lags and time features, none of them tackles the problem by learning an autoregressive model in the expressive space of time-localized frequencies.

## 3 LANGUAGE MODELING OF TIME-LOCALIZED FREQUENCIES

We introduce a wavelet-based tokenizer that allows the model to learn complex representations directly in the space of time-localized frequencies. Our method first scales and decomposes the input time series, then thresholds and quantizes the wavelet coefficients, and finally it pre-trains an autoregressive model to forecast wavelet coefficients for the horizon window.

### 3.1 A BRIEF TOUR OF WAVELETS

Here we provide a very brief introduction to the main concepts and terminology regarding wavelets. See Appendix A and references therein for more details. The Wavelet transform (WT) was intro-

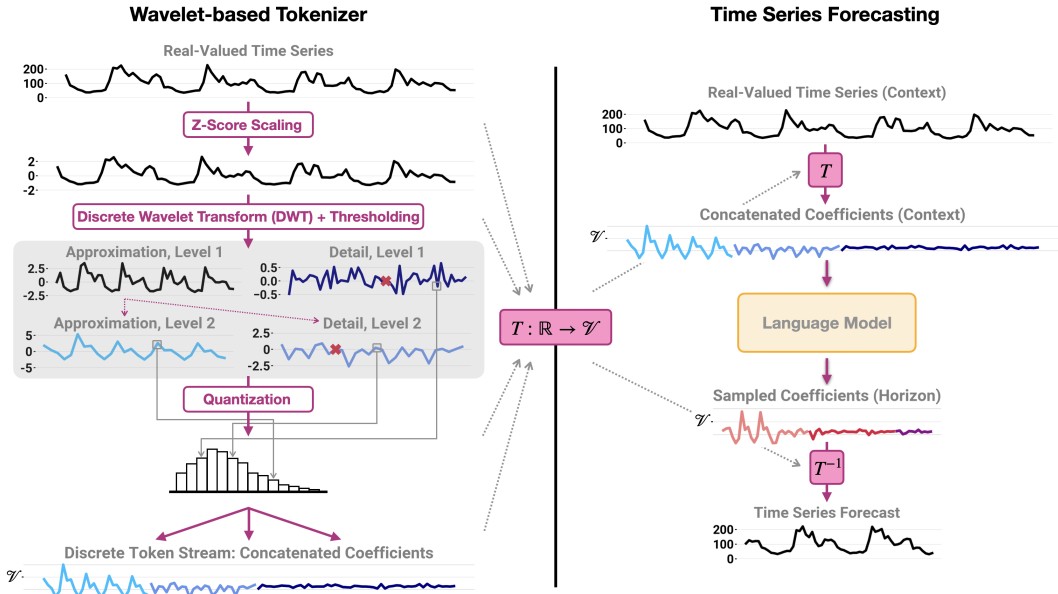

Figure 2: **High-level depiction of our method.** *(Left)* WaveToken first re-scales the input time series by computing $\tilde{x}_t = (x_t - \mu_{1:C})/\sigma_{1:C}$, then it applies the DWT and possibly thresholds the resulting detail coefficients to zero (red crosses). The wavelet coefficients are finally quantized to bins of optimal size given their empirical distribution, and then concatenated together (excluding the first $J-1$ approximations). *(Right)* After pretraining on a large corpus of time series, at inference time the model samples autoregressively from the categorical output distribution and yields coefficients at all decomposition levels, which are pushed through the inverse tokenizer to obtain a forecast.

duced to address some limitations in existing mathematical tools — namely the Fourier transform, which implicitly assumes a signal is stationary — by employing a quickly decaying zero-mean oscillatory function, known as the mother wavelet, which inherently adapts its time-frequency resolution to the signal's characteristics. This dual localization property, achieved through the modulation of the wavelet's scale parameter, enables the WT to efficiently analyze non-stationary signals achieving localization in both time and frequency domain (Daubechies, 1992; Mallat, 2009).

We can divide wavelet families into two sets of basis functions: the father wavelet, which captures low frequencies (the *approximation*), and the mother wavelet, which focuses on high frequencies (the *detail*). Both sets can be modulated via scaling and translation to achieve a multi-resolution decomposition of an arbitrary signal $f(x)$ into the $J$-th lowest approximation component combined with the $J-1$ successive details. In what follows, we obtain approximation $\{a_k\}_J$ and detail coefficients $\{d_k\}_j$, $\forall k, j$ via the maximally decimated Discrete Wavelet Transform (DWT), which decomposes a signal and preserves its length $N$ by applying a cascade of high-pass and low-pass conjugate mirror filters via successive convolutions and down-sampling operations in $\mathcal{O}(N)$ time.

## 3.2    TOKENIZATION VIA WAVELET DECOMPOSITION

We ask the following question: given a real-valued univariate time series $\mathbf{x}_{1:C} = [x_1, \ldots, x_C]$, where $C$ is the context length, can we find an optimal map $T : \mathbb{R} \to \mathcal{V}$ that encodes the input with a compact but expressive *discrete* vocabulary $\mathcal{V}$? To this end, we develop WaveToken, a tokenization pipeline divided in scaling, wavelet decomposition, thresholding and quantization. See Section 4.4 for an extensive analysis of the chosen hyper-parameters.

**Scaling.**    Re-normalizing time series is standard practice in modern forecasting, as it brings all inputs to a common scale and avoids numerical and optimization issues, especially for globally-learned deep learning models (Benidis et al., 2022). Popular normalization techniques are based on affine transformations $\tilde{x}_t = (x_t - m)/s$, with $m$ and $s$ appropriately chosen. We set $m =$

$\mu_{1:C} = \frac{1}{C}\sum_{t=1}^{C} x_t$ and $s = \sigma_{1:C} = \sqrt{\frac{1}{C-1}\sum_{t=1}^{C}(x_t - \mu_{1:C})^2}$, also known as z-score scaling. This choice is especially important in our case because the specific wavelet transform we adopt is not translation-invariant, hence a shift in the input signal can lead to different coefficients. Other popular normalization techniques — e.g., dividing by the mean of the absolute values in the context — could therefore yield unexpected results for similar inputs. By applying this transformation, we make sure the model receives consistent time-frequency representations.

**Decomposition.** We then apply the *maximally decimated* DWT (Daubechies, 1992) to decompose the scaled time series $\tilde{\mathbf{x}}_{1:C}$ into its constituent time-localized frequencies up to the $J$-th level, which yields approximation coefficients $\{a_k\}_J$ and a series of detail coefficients $\{d_k\}_{j=1}^{J}$, $\forall k$. In addition to being faster than the FFT as mentioned in Section A.2, this wavelet transform leads to compact representations since it preserves the input size — i.e., a signal of size $N$ results in $N$ coefficients[1] — and tends to concentrate the majority of the signal energy onto only a few significant wavelet coefficients. This is especially true for time series with sharp spikes or localized features, and is a useful property for compression and denoising. As each coefficient group is the outcome of convoluting a filter with the signal, the autoregressive structure is also preserved *within* each group.

**Thresholding.** The inherently sparse representations that the DWT induces imply that we can encode complex features with only a few significant wavelet coefficients, while the rest are close to zero and thus can potentially be discarded without significantly addressing reconstruction quality. Several thresholding techniques, many tailored to specific applications, have been developed over the years and are effectively used in practice (e.g., in image compression: Chang et al. (2000); Vetterli & Kovacevic (1995); Christopoulos et al. (2000)). As the main downstream task in this work is to pre-train a language model on a large corpus of diverse time series datasets, any thresholding technique must be adaptive to the underlying noise and complexity level of the input, so as to retain most of the signal energy without discarding essential information. In what follows, we always keep the approximation coefficients unaltered since they model the low-resolution coarse structure of the time series, and explore the following techniques for thresholding detail coefficients:

*No-thresholding*: $\bar{d}_{k,j} = d_{k,j}$. If the chosen wavelet family preserves signal energy (in the sense of Parseval's theorem), not thresholding the coefficients avoids any information loss at this stage.

*CDF-thresholding*: $\bar{d}_{k,j} = d_{k,j}\mathbf{1}\{|d_{k,j}| > F_{|d_j|}^{-1}(b^{J-j+1})\}$. A coefficient is set to 0 if it is in the lower tail of the empirical distribution of the details' magnitude at the corresponding $j$-th level, where the cutoff grows exponentially from coarser to finer coefficients to reflect granularity and downsampling of the DWT.

*VisuShrink (Donoho, 1995)*: $\bar{d}_{k,j} = \text{sign}(d_{k,j})(|d_{k,j}| - \lambda)$, with $\lambda = \sigma\sqrt{2\log N}$ and $\sigma$ estimated from the finest detail coefficients $d_{k,j=1}$. The DWT of noisy data can be seen as a maximum-likelihood estimate of the wavelet coefficients, and it can be shown that this threshold reduces the expected reconstruction error (i.e., estimator's risk) close to the possible minimum, under certain assumptions. We explore both the soft- and hard-thresholding variants of this method.

*FDRC (Abramovich & Benjamini, 1996)*: see Appendix B for details on the algorithm. By leveraging the connection between thresholding and multiple hypotheses testing, this procedure improves on *VisuShrink* by adaptively choosing $\lambda$ to control the expected proportion of incorrectly included coefficients (similar to false discovery rate) among those chosen for the wavelet reconstruction.

See Section 4.4 and Appendix B for results and justifications on the chosen thresholding technique.

**Quantization.** At this point, the resulting coefficients $\{a_k\}_J$ and $\{\bar{d}_k\}_{j=1}^{J}$ are still real-valued and need to be converted into discrete tokens to be directly processed by language models. We construct the vocabulary $\mathcal{V}$ by binning the raw coefficients according to their joint empirical distribution on the training set, and choose the optimal bin size according to the Freedman-Diaconis rule (FD) to minimize the reconstruction error (Freedman & Diaconis, 1981). In symbols, given the optimal $B$ bin centers $c_1 < \cdots < c_B$ and $B - 1$ edges $c_i < e_i < c_{i+1}$ for $i = 1, \ldots, B - 1$, we map each wavelet coefficient $w \in \{\{a_k\}_J, \{\bar{d}_k\}_j\}$ to $\mathcal{V}$ as follows: $q(w) = i \cdot \mathbf{1}\{e_{i-1} \leq w < e_i\}$. We

---

[1]To be precise, longer filters and specific boundary conditions to handle the signal edges might yield a *slightly* higher number of coefficients (Torrence & Compo, 1998).

further enrich $\mathcal{V}$ to be immediately compatible with language models by adding two special tokens: a PAD token that signals missing values in the wavelet coefficients — resulting from missing values in the time series or from padding of short instances — and an EOS token that signals the end of the sequence. In our context, a single shared vocabulary jointly encodes time-localized low and high frequencies, thereby leading to a compressed but expressive codebook.

### 3.3 MODEL TRAINING, OBJECTIVE FUNCTION AND FORECASTING

Given a time series context $\mathbf{x}_{1:C}$, we apply the steps detailed in Section 3.2 and then concatenate the resulting discrete tokens for approximation and detail coefficients into a vector $\mathbf{z}_{1:C} = [\mathbf{a}_J, \mathbf{d}_J, \dots, \mathbf{d}_1]$, so that the model can learn and forecast coarse-to-fine structures in a multi-scale fashion. To minimize ad-hoc modifications to the language model, we opt for an encoder-decoder architecture based on the T5 family (Raffel et al., 2020), which has recently been shown to achieve excellent zero-shot performance on a comprehensive benchmark (Ansari et al., 2024). Other proposed LLM-based forecasters either reformulate the problem as a question-answering task or use patches of the input as tokens, and would therefore not be immediately compatible with our wavelet-based tokenizer (see section 2 for an exhaustive review). For brevity, in what follows we refer to WaveToken as being both the tokenizer and the model paired together.

We train the model via next-token prediction by minimizing the cross-entropy between the predicted distribution and the categorical output distribution over the corresponding label tokens obtained from the horizon window $\mathbf{z}_{C:C+H}$. In symbols, the loss function for a time series (including EOS) is

$$l_\theta = -\sum_{h=1}^{H+1} \sum_{i=1}^{|\mathcal{V}|} \mathbf{1}\{z_{C+h+1} = i\} \log p_\theta(z_{C+h+1} = i \mid \mathbf{z}_{1:C+h})$$

While the cross-entropy loss does not induce a metric onto the underlying vocabulary, it offers greater flexibility to learn output distributions of arbitrary shapes. This property lends itself well to forecasting applications, where it is often important to capture the correct shape or pattern of the time series without imposing structural limitations such as those inherent in traditional loss functions like MSE and MAE (Le Guen & Thome, 2019).

Learning an autoregressive model on concatenated groups of wavelet coefficients might seem counter-intuitive: the temporal structure is only preserved within each coefficient group and is broken as the input transitions from, e.g., approximations to details. In practice, this turns out to be surprisingly helpful as it offers the model a natural way to break down complex sub-structures in inputs and outputs by dividing them into a hierarchy of time-localized frequency bands. As we will see in Section 4.3, the model easily learns to exploit these partitions and is able to attend to the right coefficients to improve the overall forecasting accuracy.

At inference time, the model produces sample paths over the vocabulary via autoregressive sampling from the predicted distribution $p_\theta(z_{C+h+1} \mid \mathbf{z}_{1:C+h})$, $h = 1, \dots, H$. To obtain a time series forecast, we first de-quantize the tokens by mapping them to the corresponding bin center: $q^{-1}(i) = c_i$. We then apply the inverse discrete wavelet transform (IDWT) and un-scale the reconstructed series by multiplying it by $\sigma_{1:C}$ and adding $\mu_{1:C}$, as depicted in Figure 2.

## 4 EXPERIMENTS

### 4.1 SETUP OF EMPIRICAL EVALUATION

**Models and baselines.** We pre-train WaveToken with T5 models of four sizes[2] — Mini (19.2M), Small (44.5M), Base (199M) and Large (705.8M) — for 200K steps on 8 A100 GPUs, and we compare their performance against *i)* popular task-specific models trained for each dataset separately, namely DeepAR (Salinas et al., 2020), PatchTST (Nie et al., 2022) and TFT (Lim et al., 2021); and *ii)* recently proposed foundation models for time series forecasting — namely, TimesFM (Das et al., 2023), Chronos Mini-Large (Ansari et al., 2024), Moirai Base & Large (Woo et al., 2024), and Lag-llama (Rasul et al., 2023) — which do not perform task-specific training, but are trained only

---

[2]Parameter counts were obtained with the optimal vocabulary size as detailed in Section 4.4.

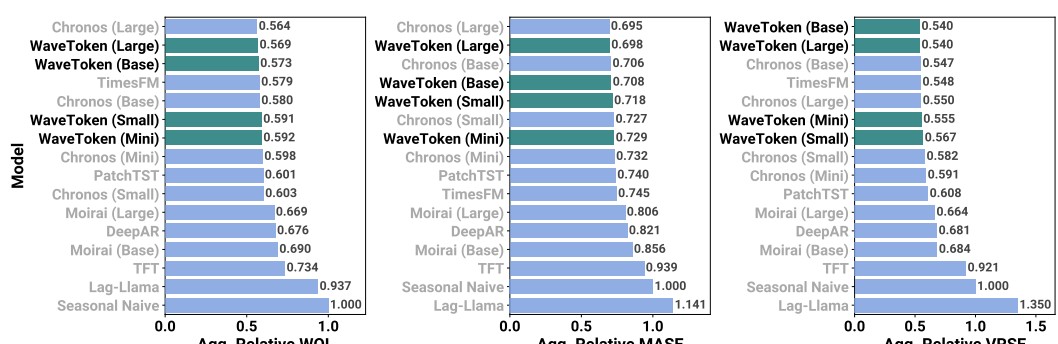

Figure 3: **WaveToken performs on par or better than other baselines on in-domain datasets.** Forecasting accuracy on Benchmark I in terms of WQL, MASE and VRSE.

once on a large corpus and then deployed across all evaluation datasets. Appendix G details all the hyper-parameters for the mentioned baselines.

**Datasets and training strategy.** We train and evaluate WaveToken on the publicly available datasets comprehensively collected by Ansari et al. (2024). These span a variety of domains and exhibit diverse properties in terms of size, frequencies and prediction lengths, and can be divided in *i) pre-training only*: datasets exclusively used for training (13 datasets); *ii) in-domain*: datasets employed for training, whose validation set is also used for evaluation (Benchmark I, 15 datasets); and *iii) zero-shot*: datasets used solely for evaluation (Benchmark II, 27 datasets). See Appendix G for detailed information about all datasets. The context length of the sequences in each training batch is set to 512 and the prediction length is set to 64. In addition, we adopt the data augmentations techniques introduced by Ansari et al. (2024): each of the sequences is generated with probability 0.9 from a TSMixup set, which takes convex combinations of different time series, and with probability 0.1 from a synthetic dataset generated from Gaussian Processes based on randomly combined kernels. Both of these have been shown to be effective in domains such as time series forecasting, where data is inherently scarce relative to standard language modeling applications.

**Evaluation tasks and metrics.** For both in-domain (I) and zero-shot (II) benchmark datasets, we use the last $H$ observations of each time series as a held-out test set. We compute the weighted quantile loss (WQL) to assess the quality of probabilistic forecasts on 9 uniformly-spaced quantile levels $\{0.1, 0.2, ..., 0.9\}$ and the mean absolute scaled error (MASE; Hyndman & Koehler (2006)) to evaluate the quality of point forecasts. In addition, we compute the Visual Relative Squared Error (VRSE; Posam et al. (2024)), which measures the relative squared difference of the amplitudes at all frequencies between the ground truth and the (median) forecast. This metric serves a complementary role to avoid common pitfalls of standard scores, which can fail to properly assess important edge cases that would otherwise be visually obvious, as shown in Figure 9. In order to aggregate these metrics and provide fair comparisons, we compute each model's score divided by the score of a baseline model (here, Seasonal Naive). These relative scores are then aggregated across all datasets using the geometric mean. See Appendix C for more details. All results for Chronos and WaveToken are averaged across three different seeds.

## 4.2 In-Domain & Zero-Shot Benchmarks

Figure 3 summarizes the forecasting performance of all models on the in-domain datasets of Benchmark I. WaveToken outperforms all other baselines with the exception of Chronos-Large on WQL and MASE, and is superior with respect to VRSE. Considering each model size for each metric, WaveToken achieves lower (i.e., better) scores than Chronos 75% of the times and largely improves on other recent LLM-based forecasters. Similarly, our method is considerably better even than task-specific models trained separately on each dataset. Note again that WaveToken uses a vocabulary size of 1024, while Chronos, for example, uses four times as much tokens with $|\mathcal{V}| = 4096$. This empirical finding confirms the well-known theoretical properties of wavelets, which can provide compact but very expressive representations able to condense complex structures into a compressed

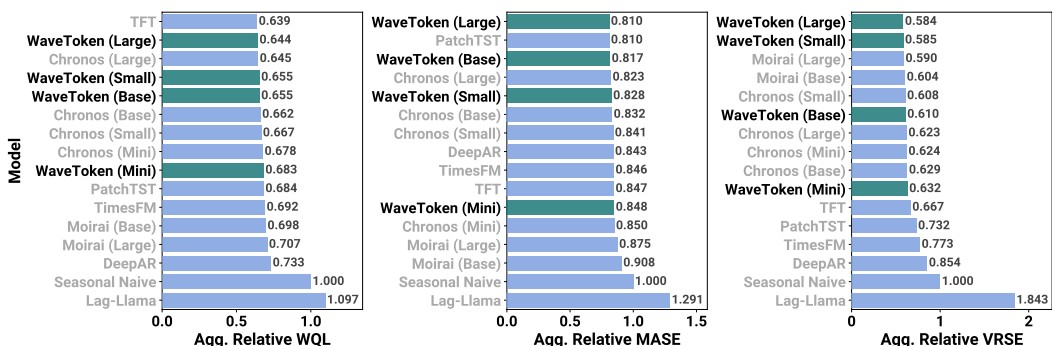

Figure 4: **WaveToken performs on par or better even relative to task-specific models on zero-shot datasets.** Forecasting accuracy on Benchmark II in terms of WQL, MASE and VRSE.

codebook. Finally, on the datasets of Benchmark I, WaveToken achieves the best average rank across all metrics, as shown in Figure 10.

Benchmark II focuses on the forecasting performance on datasets that were never included in the training corpus (for brevity, zero-shot). As shown in Figure 4, WaveToken exhibits superior generalization capabilities which lead it to *i)* outperform all other foundation models for time series across all metrics, with a 83% success rate against Chronos models of the same size; and *ii)* be competitive on WQL and MASE, and much better on VRSE, relative to task-specific models specifically trained on each zero-shot dataset. Again, WaveToken achieves the best average rank across all metrics on the dataset of Benchmark II, as shown in Figure 11.

Figure 12 in Appendix E shows results on a long-horizon benchmark constructed by increasing the forecast length $H$ of each dataset in Benchmark II (Zero-Shot) by a factor of 2 and 3. WaveToken outperforms other foundation models across almost all combinations of metrics and long horizons, the only exception being $H \times 3$ with respect to WQL, where TimesFM performs slightly better. Appendix D shows the raw WQL, MASE and VRSE values for each dataset in Benchmark I and II.

### 4.3 QUALITATIVE ANALYSIS

So far we have seen how wavelets allow a transformer-based architecture trained on a large corpus to achieve excellent forecasting performance especially on previously unseen datasets, which hints at their superior generalization capabilities. It is then worth analyzing more deeply some edge cases of practical relevance to showcase how wavelets can efficiently capture a wide variety of complex patterns, frequencies and local structures within a compressed representation. Figure 1 shows examples of synthetically-generated time series which exhibit strong trends, sharp spikes and several frequencies evolving over time. We evaluate WaveToken against popular recent foundation models for time series forecasting. Both Chronos, TimesFM and Moirai clearly underestimate trends, struggle at isolating sudden spikes and are not able to capture non-stationary behaviours. On the other hand, our model is able to leverage the different concatenated coefficient groups which represent the time-localized frequency bands, thereby providing accurate forecasts with very low uncertainty.

This phenomenon can be further explained by looking at patterns in the cross-attention layers of Chronos and our model, as they use the same T5 encoder-decoder architectures. Figure 5 shows heat-maps of the cross-attention weights in the eighth decoder layer of Chronos-Base (left) and WaveToken-Base (right), when forecasting the sparse spikes in the second row of Figure 1. Two main things are worth noticing: first of all the attention map for our model is clearly divided in four quadrants, of which the upper-left and lower-right ones exhibit generally larger attention values. These clearly show that, to forecast approximation coefficients in the horizon (time-steps 0-34 on the $x$-axis), the model is learning to attend more to approximation coefficients in the context (time-steps 0-258 on the $y$-axis), and to forecast detail coefficients in the future (time-steps 35-68 on the $x$-axis), the model attends more to detail coefficients in the context (time-steps 259-516 on the $y$-axis). Second, we notice an interesting pattern: the detail coefficients for the horizon present two

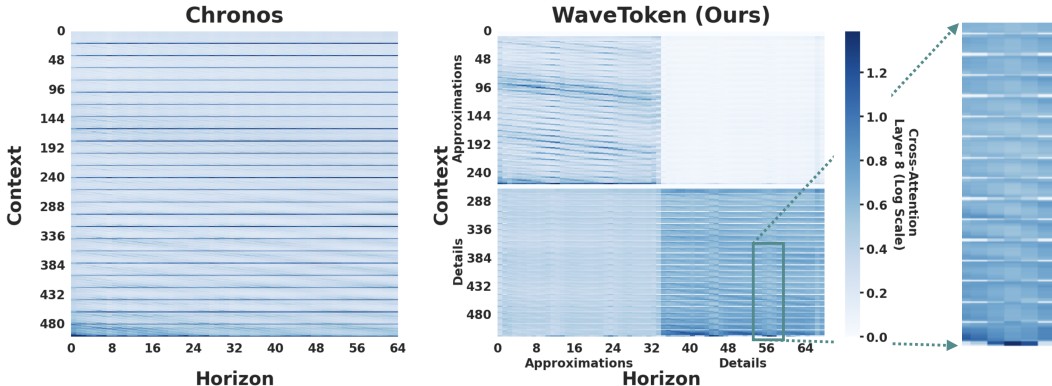

Figure 5: **Wavelet-based tokenization induces structured patterns in the cross-attention maps.** Cross-attention weights for the eighth decoder layer when forecasting the spiky data of Figure 1 (second row). Chronos-Base (left) repeats the same patterns for all steps, while `WaveToken-Base` (right) shows high values at the detail coefficients corresponding to the spikes.

clear columns of larger attention weights (time-steps $45$ and $56$ on the $x$-axis) that map to the detail coefficients representing the spikes in the context. In addition, other detail coefficients attend to everywhere else in the bottom-right quadrant (corresponding to the flat regions in time domain for the spiky data of Figure 1) except to those same positions corresponding to spikes (see the white horizontal lines interrupted by the vertical column in the magnified portion). Similar patterns are present for the approximation coefficients, albeit less defined. None of this structures is present in the attention weights of Chronos-Base shown in the left panel, which repeats the same patterns for all steps and indeed fails to forecast the correct position and intensity of the spikes in time domain.

Appendix F shows the remaining attention maps for layers missing in Figure 5, which all show similar patterns with varying intensities.

### 4.4 ABLATION STUDY

Figure 6 shows the effect of different vocabulary sizes, wavelet families and decomposition levels on the forecasting accuracy of `WaveToken-Small` trained for 200K steps on one A100 GPU. Regarding *vocabulary size*, which determines the precision with which tokens encode wavelet coefficients, we observe a gradual but consistent improvement until $|\mathcal{V}| = 1024$, which we select as the optimal value. For higher vocabulary sizes, WQL and MASE remain flat or worsen on both in-domain and zero-shot benchmarks. This phenomenon can be ascribed to the intrinsic compression properties of wavelets (Mallat, 2009): by concentrating most of the signal energy onto a few coefficients, we can effectively capture more information with a smaller codebook, while a larger one would only reserve more tokens to spurious coefficients. Note that our vocabulary is much smaller than comparable models: Chronos for example, that leverages the same architectures, uses $|\mathcal{V}| = 4096$. As to the many *wavelet families* available, the central panel of Figure 6 shows that the `Biorthogonal-2.2` basis achieves optimal performance. This family uses two separate filters with two vanishing moments for analysis and synthesis, a dual structure that allows for symmetric or near-symmetric wavelet functions that help preventing distortions near the signal edges. This is known to be advantageous in many applications, including image compression[3]. Families with higher vanishing moments capable of capturing higher order polynomial did not prove effective. As to the *decomposition level*, we observe that first-level coefficients are sufficient to achieve good forecasting accuracy. Albeit deeper levels provide more granular time-localized frequencies, the higher number of groups in the concatenated coefficients make it harder for the attention mechanism to identify the relevant ones at each prediction step. Regarding *thresholding*, we analyze the performance of all four techniques described in Section 3. Empirically, we observe a discrepancy in the results obtained when training

---

[3]Biorthogonal wavelets with the maximally decimated DWT are in fact used in popular compression standards, such as JPEG-2000 (Christopoulos et al., 2000; Usevitch, 2001).

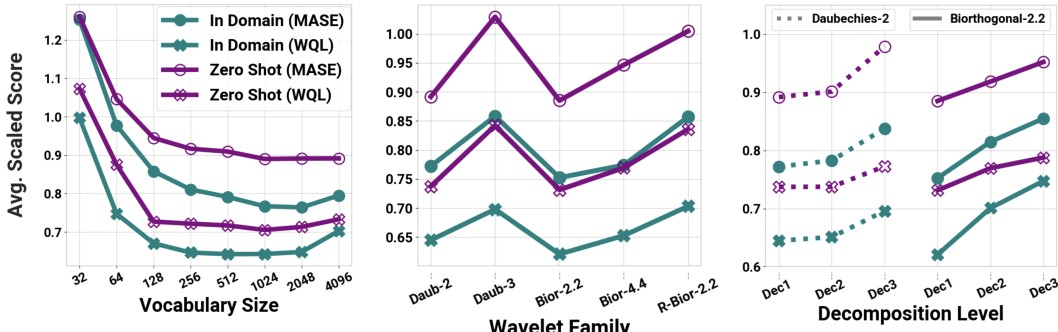

Figure 6: **Effect of different vocabulary sizes, wavelet families and decomposition levels on downstream forecasting accuracy of** `WaveToken-Small`**.** The optimal hyper-parameters were a single-level wavelet decomposition with a `Biorthogonal-2.2` family and a vocabulary size of 1024. See Section 4 for more details.

for the same number of steps on 1 or 8 GPUs, where the latter is the final configuration used to train the optimal models. As Figure 8 shows, in the single-GPU setting, chosen to streamline computations during hyper-parameter optimization, *VisuShrink* (Donoho, 1995) appears to be the best, a result which is then reverted in favor of *No-thresholding* in the 8-GPU setting, which increases the total number of samples processed at each step, thereby allowing the model to learn the richer frequencies preserved in the input. *VisuShrink*, on the other hand, leads to smoother signals by crudely cutting off coefficients, which eventually harms generalization. These empirical observations led us to not threshold coefficients, although we note that thresholding is still implicitly happening during the *quantization* step, which collapses small coefficients to the bin centered at 0. This value is then used at inference time to forecast tokens mapped to this bin. Optimal bins for the discretization procedure — detailed in Section 3 — were selected between the lower and upper bounds $[-30, 30]$, chosen empirically by scanning the training corpus.

## 5 CONCLUSION

In this work, we develop `WaveToken`, a tokenization pipeline tailored to a specific goal: constructing a general-purpose forecasting model capable of capturing a wide variety of complex patterns while consuming as little information as possible, thereby leading to excellent generalization performance on unseen datasets. We leverage a recently-proposed framework to pre-train an encoder-decoder architecture in the context of time series (Ansari et al., 2024) and re-purpose it to learn an autoregressive model in the space of time-localized frequencies. The resulting wavelet-based vocabulary is both compact — using 1024 tokens, i.e. one quarter of Chronos — and very expressive, and leads to

*i)* excellent **forecasting accuracy** with respect to all other baselines in terms of three complementary metrics: weighted quantile loss for probabilistic forecasts, mean absolute scaled error for point forecasts, and visual relative squared error to measure dicrepancies in the frequency content relative to the ground truth.

*ii)* superior **generalization** capabilities, with `WaveToken` being the best model across all datasets and metrics in terms of average rank. In addition, our method can easily capture complex temporal patterns in several edge cases relevant for practical applications, as shown in Figure 1.

As potential future directions, we foresee that exploring techniques to automatically handle context lengths larger than 512 in the tokenizer would allow models to capture longer-range dependencies if needed. The DWT is in fact a natural tool to leverage in these settings, as it can encode long contexts without increasing the input length through convolutions and down-sampling. Furthermore, `WaveToken` exploits an autoregressive model on wavelet coefficients. As such, it suffers from slower decoding and inference times relative to recently proposed alternatives, such as patch-based models. Applying `WaveToken` to patch-based architectures represents an interesting area for future research.

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

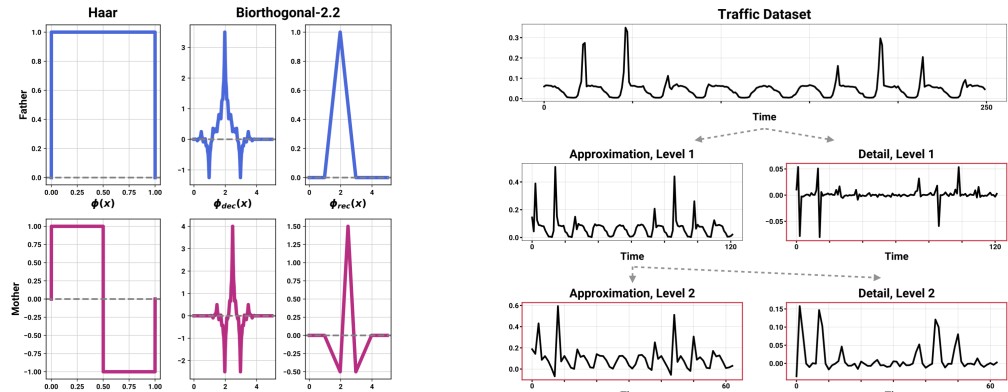

Figure 7: *(Left)* Mother and father wavelets for the `Haar` and `Biorthogonal-2.2` families. Note the dual structure of the latter, which uses two filters: one for decomposition and one for reconstruction. *(Right)* Example of discrete wavelet transform (DWT) applied to a time series from the traffic dataset. Red boxes highlight the coefficients that are returned by the decomposition.

## A  A BRIEF TOUR OF WAVELETS

### A.1  MOTIVATION: FROM THE FOURIER TRANSFORM TO THE WAVELET TRANSFORM

The Fourier Transform (FT) allows one to map a signal from the time domain to the frequency domain by computing the signal's projection onto a basis of complex exponentials, which represent sine and cosine functions of varying frequencies. This process effectively decomposes the signal into its constituent frequency components. The main drawback of the Fourier Transform is that, while having high resolution in the frequency domain, it has zero resolution in the time domain. In other words, it cannot tell at which location in time these frequencies occur in the original signal. The Short-Time Fourier Transform (STFT) tries to overcome this issue by splitting the original signal in different windows and applying the Fourier Transform in each of them. Nonetheless, the fixed window immediately implies a trade-off: the smaller the size of the window, the higher the resolution in time domain, the lower the resolution in frequency domain, and vice versa. This is widely known as the Uncertainty Principle (Oppenheim & Schafer, 2010).

The Wavelet Transform (WT) addresses this limitation by employing a quickly decaying zero-mean oscillatory function, known as the mother wavelet, which inherently adapts its time-frequency resolution to the signal's characteristics. By dilating or compressing the wavelet, its time support varies inversely with its frequency, thus providing high time resolution for high-frequency components and high frequency resolution for low-frequency components. This dual localization property, achieved through the modulation of the wavelet's scale parameter, enables the WT to efficiently analyze non-stationary signals whose spectral content evolves over time, a phenomenon often referred to as *time-frequency localization* (Daubechies, 1992; Mallat, 2009).

### A.2  THE HAAR WAVELET AND THE DISCRETE WAVELET TRANSFORM

Wavelet families consist of basis functions that can be divided into two primary types: the scaling function (often referred to as the "father" wavelet), which captures the low-frequency, coarse structure of the signal (the *approximation*), and the wavelet function (often referred to as the "mother" wavelet), which captures the high-frequency components (the *detail*) of the signal. The "father" and "mother" wavelets give rise to "son" and "daughter" wavelets through scaling and translation. Here, we briefly formalize these concepts in the context of the Haar wavelet (Haar, 1911), a simple yet pedagogically significant wavelet family. For an extensive introduction to wavelets and their applications, see, for example, Mallat (2009), Daubechies (1992) and Strang & Nguyen (1996).

The Haar father wavelet is given by $\phi(x) = \mathbb{I}(x \in [0, 1])$. The corresponding son wavelet is

$$\phi_{k,j}(x) = 2^{j/2}\phi(2^j x - k) = \begin{cases} 2^{j/2}, & \text{if } \frac{k}{2^j} \leq x \leq \frac{k+1}{2^j} \\ 0, & \text{otherwise,} \end{cases}$$

---

**Algorithm 1** False Discovery Rate of Coefficients (FDRC)

---

1: For each $d_{k,j}$, compute two-sided p-value $p_{k,j} = 2(1 - \Phi(|d_{k,j}|/\sigma))$ for $H_0^{(k,j)} : d_{k,j} = 0$
2: Order $p_{k,j}$ such that $p_{(1)} \leq \cdots \leq p_m$
3: Let $i_0 = \arg\max_i p_{(i)} \leq (i/m)q$, with $q$ being error-rate under $H_0$
4: Let $\lambda_{i_0} = \sigma\Phi^{-1}(1 - p_{i_0}/2)$
5: Threshold all detail coefficients at level $\lambda_{i_0}$

---

where $k, j \in \mathbb{N}$. Note that $\phi_{k,0}(x)$ generates an orthonormal basis for functions with jumps at the integers, whose space we denote by $V_0$. The dilations and translations $\phi_{k,j}(x)$ span orthonormal spaces $V_j$ for functions with jumps at $k/2^j$, and satisfy $V_{j+1} \supset V_j \supset \cdots \supset V_0$. With these bases, we can represent an arbitrary function $f(x)$ in $V_{j+1}$ by taking a component in $V_j$ (the *approximation*) plus the orthogonal complement of $V_j$ to $V_{j+1}$, which we denote by $W_j$ (the *detail*). In other words, we represent $f$ in $V_{j+1}$ by taking its orthogonal decomposition $V_{j+1} = V_j \oplus W_j$. The space $W_0$ is spanned by the mother wavelet

$$\psi(x) = \phi(2x) - \phi(2x-1) = \begin{cases} 1, & \text{if } 0 \leq x < \frac{1}{2} \\ -1, & \text{if } \frac{1}{2} \leq x \leq 1 \\ 0, & \text{otherwise} \end{cases},$$

whose dilations and translations $\psi_{k,j} = 2^{j/2}\psi(2^j x - k)$ form an orthonormal basis for each $W_j$. Putting this all together, we can represent a function at different resolution levels by breaking down the level-$j$ approximation component $V_j$ to level-$(j-1)$ detail and approximation components, and so on: $V_J = V_J \oplus W_J = V_0 \oplus W_0 \oplus W_1 \cdots W_{J-1}$. A wavelet decomposition is then a linear combination of elements in these subspaces:

$$f(x) = \sum_{k=-\infty}^{\infty} a_k\phi_{k,J}(x) + \sum_{j=1}^{J} \sum_{k=-\infty}^{\infty} d_{k,j}\psi_{k,j}(x),$$

where only a finite number of coefficients $a_k$ and $d_{k,j}$ are non-zero.

How do we compute $a_k$ and $d_{k,j}$ in practice? The multi-resolution theory of Mallat (1989) and Meyer (1992) offers an elegant and efficient solution: any wavelet that generates an orthonormal basis can be characterized by a conjugate mirror filter. The mapping from a discretized signal to a sequence of wavelet coefficients is then implemented as a filter bank made of a cascade of high-pass and low-pass filters associated with the mother and father wavelets, and is called Discrete Wavelet Transform (DWT)[4]. For the Haar family, the approximation components are associated with averaging filters (low-pass), while the detail components are associated with distance filters (high-pass). The DWT convolves these filters with the signal and then downsamples by a factor of two to eliminate the repeated information. The result is an array of detail coefficients $\{d_k\}_j$ and approximation coefficients $\{a_k\}_j$. The latter can be further decomposed into new approximation and detail coefficients at the next level and the process can continue recursively, resulting in a hierarchical and multi-resolution decomposition of the signal. Figure 7 (right) shows this for a particular time series.

The DWT has a computational complexity of $\mathcal{O}(N)$, with $N$ being the length of the input signal. This is due to *i)* the filtering operation requiring a constant amount of work proportional to the signal length, and *ii)* the down-sampling operation halving the signal at each of the $\log_2 N$ levels: this leads to a total amount of work equal to $N(1 + \frac{1}{2} + \frac{1}{4} + \cdots)$, which converges to $2N$. Note that this is even faster than the FFT algorithm, which has a computational complexity of $\mathcal{O}(N \log N)$.

## B    ADDITIONAL RESULTS ON DIFFERENT THRESHOLDING TECHNIQUES

Algorithm 1 outlines the FDRC thresholding method step-by-step. See Abramovich & Benjamini (1996) for more details. During our experiments, we chose the standard $q = 0.05$, which corresponds to the type-I error control level of the resulting hypotheses tests.

Figure 8 shows the effect of the different thresholding techniques detailed in Section 3.2 on the downstream forecasting accuracy in terms of weighted quantile loss (WQL), mean absolute scaled

---

[4]We always refer to the *decimated* DWT. For a comparison with the *undecimated* DWT, see Mallat (2009).

error (MASE), and visual relative squared error (VRSE). This ablation study complements those on vocabulary size, wavelet family and decomposition level outlines in Section 4.4. Apart from the four thresholding techniques analyzed in this paper, several more exist in the literature for signal processing and image compression. We leave a comprehensive analysis of these methods and a deeper study of their effect on downstream performance to future work.

## C  EVALUATION METRICS

Consider a collection of $N$ time series $\{\mathbf{x}_i = [x_{i,1}, \ldots, x_{i,C+H}]\}_{i=1}^N$ that include both context and horizon. In Section 4, we evaluated WaveToken and all the other baselines with respect to three metrics, which we now describe more in detail: weighted quantile loss (WQL), mean absolute scaled error (MASE), and visual relative squared error (VRSE)

**Weighted quantile loss (WQL).**  We use this metric to evaluate probabilistic forecasts $q_{i,C+t}^\alpha$ — obtained by generating $N = 20$ samples from the model for an input $i$ — at nine quantile levels $\alpha \in \{0.1, 0.2, 0.3, 0.4, 0.5, 0.6, 0.7, 0.8, 0.9\}$. WQL aggregates the standard quantile loss at level $\alpha$ $\mathrm{QL}_\alpha(q,x)$ (Koenker & Hallock, 2001) over multiple horizon steps $t = 1, \ldots, H$ and series $i$ by taking a weighted average:

$$\mathrm{WQL} = 1/9 \sum_\alpha \frac{2 \sum_{i,t} \mathrm{QL}_\alpha(q_{i,C+t}^\alpha, x_{i,t})}{\sum_{i,t} |x_{i,t}|},$$

where 9 is the number of quantiles used.

**Mean absolute scaled error (MASE).**  We use this metric to evaluate point forecasts $\hat{\mathbf{x}}_i = [x_{i,1}, \ldots, x_{i,C+H}]$, which we take to be the median quantile $q_{i,C+t}^{0.5}$ across $N = 20$ samples for probabilistic models. The MASE (Hyndman & Koehler, 2006) scales the mean absolute error by the empirical error of the seasonal naïve model:

$$\mathrm{MASE}(\hat{\mathbf{x}}_i, \mathbf{x}_i) = \frac{C - S}{H} \frac{\sum_{t=C+1}^{C+H} |\hat{x}_{i,t} - x_{i,t}|}{\sum_{t=1}^{C-S} |x_{i,t} - x_{i,t+S}|},$$

where $S$ is a seasonality parameter.

**Visual relative squared error (VRSE).**  We use this metric to measure the frequency content of (point) forecasts relative to the ground truth. As for MASE, we take the median forecast $q_{i,C+t}^{0.5}$ as input to the metric for probabilistic models. VRSE (Posam et al., 2024) is defined as follows:

$$\mathrm{VRSE}(\hat{\mathbf{x}}_i, \mathbf{x}_i) = \frac{\sum_f (A_{\hat{\mathbf{x}}_i}(f) - A_{\mathbf{x}_i}(f))^2}{\sum_f (A_{\mathbf{x}_i}(f))^2},$$

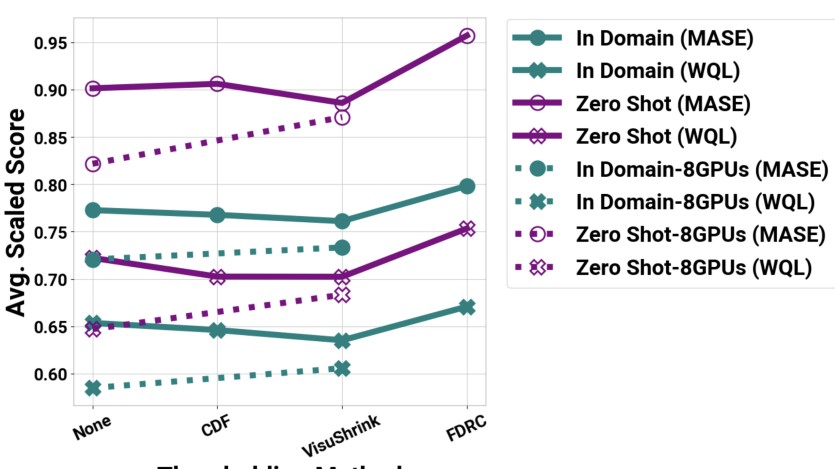

Figure 8: Effect of the different thresholding techniques of Section 3.2 on the forecasting accuracy.

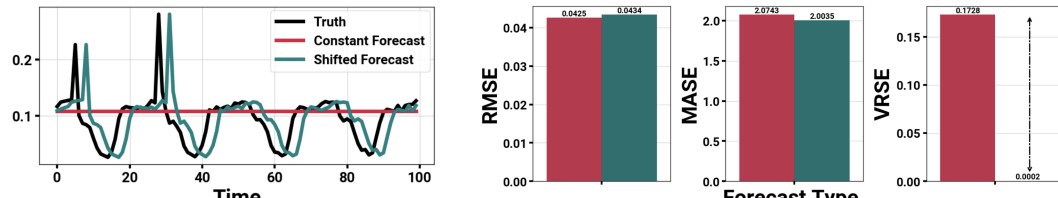

Figure 9: **Pitfall of standard evaluation metrics.** RMSE and MASE fail to distinguish a poor constant forecast from a visually much more accurate shifted forecast. By comparing the amplitudes at all frequencies, VRSE captures the difference and assigns a lower score to the shifted forecast.

where $A_x(f)$ is the amplitude of the Fourier transform coefficient at frequency $f$ for input $x$. It serves a complementary role to WQL and MASE: by computing the discrepancy between forecasts and ground truth in the amplitudes of the Fourier transform coefficients at all frequencies, this metric captures whether the forecast has the correct overall "shape", instead of looking at it point-wise. Thus, it provides a valuable alternative in cases where other metrics clearly fail, as shown in Figure 9.

## D  ADDITIONAL RESULTS: BENCHMARKS I & II

Figures 10 and 11 report the average rank across all datasets in the corresponding benchmark (in-domain and zero-shot, respectively) achieved by all the evaluated models. Tables 1-3 report the raw per-dataset values of all metrics for all models on the in-domain benchmark, along with the corresponding Aggregate Relative Score and Average Rank. Similarly, Tables 4-6 report the same values on the zero-shot benchmark. Results for WaveToken, Chronos, PatchTST, DeepAR and TFT are averaged over three seeds.

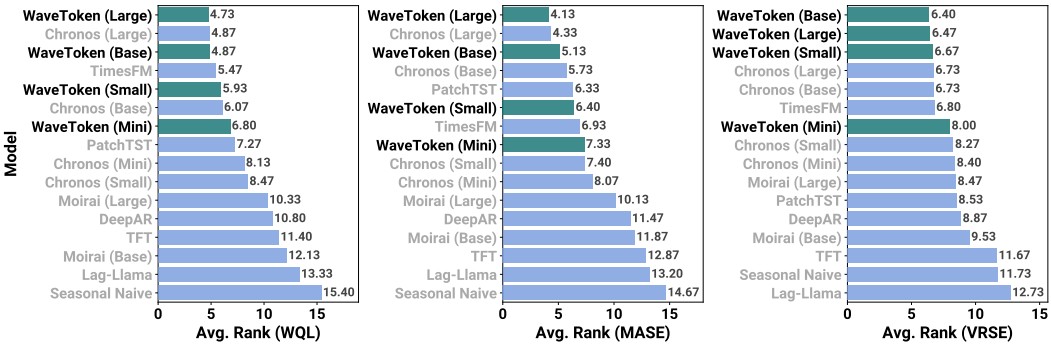

Figure 10: **WaveToken achieves best average ranks on in-domain datasets.** Average rank of models on Benchmark I (in-domain) in terms of WQL, MASE and VRSE.

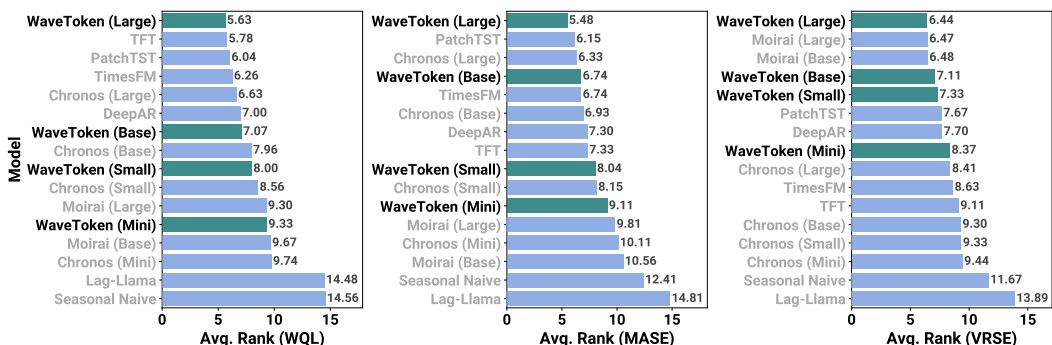

Figure 11: **WaveToken achieves best average ranks on zero-shot datasets.** Average rank of models on Benchmark II (zero-shot) in terms of WQL, MASE and VRSE.

Table 1: Raw per-dataset values of WQL for all models on the in-domain benchmark.

| | Pretrained Models (In Domain) | | | | | | | | Pretrained Models (Other) | | | | Task Specific Models | | | Local Models |
| --- | --- | --- | --- | --- | --- | --- | --- | --- | --- | --- | --- | --- | --- | --- | --- | --- |
| | Chronos (Large) | Chronos (Base) | Chronos (Small) | Chronos (Mini) | WaveToken (Mini) | WaveToken (Small) | WaveToken (Base) | WaveToken (Large) | Lag-Llama | Moirai (Base) | Moirai (Large) | TimesFM | PatchTST | DeepAR | TFT | Seasonal Naive |
| Electricity (15 Min.) | 0.077 | 0.078 | 0.080 | 0.082 | 0.086 | 0.082 | 0.079 | 0.085 | 0.319 | 0.105 | 0.104 | 0.121 | 0.082 | 0.090 | 0.189 | 0.117 |
| Electricity (Hourly) | 0.101 | 0.114 | 0.105 | 0.089 | 0.104 | 0.109 | 0.114 | 0.098 | 0.104 | 0.122 | 0.117 | 0.079 | 0.089 | 0.106 | 0.125 | 0.147 |
| Electricity (Weekly) | 0.059 | 0.062 | 0.073 | 0.067 | 0.071 | 0.069 | 0.067 | 0.065 | 0.147 | 0.113 | 0.162 | 0.062 | 0.069 | 0.116 | 0.106 | 0.198 |
| KDD Cup 2018 | 0.272 | 0.268 | 0.289 | 0.271 | 0.262 | 0.265 | 0.270 | 0.282 | 0.369 | 0.287 | 0.277 | 0.288 | 0.252 | 0.330 | 0.571 | 0.556 |
| London Smart Meters | 0.423 | 0.428 | 0.431 | 0.436 | 0.355 | 0.349 | 0.343 | 0.340 | 0.384 | 0.358 | 0.350 | 0.384 | 0.346 | 0.405 | 0.365 | 0.541 |
| M4 (Daily) | 0.022 | 0.022 | 0.022 | 0.022 | 0.021 | 0.021 | 0.021 | 0.021 | 0.043 | 0.023 | 0.023 | 0.021 | 0.023 | 0.023 | 0.023 | 0.028 |
| M4 (Hourly) | 0.022 | 0.024 | 0.024 | 0.025 | 0.025 | 0.031 | 0.024 | 0.026 | 0.111 | 0.025 | 0.022 | 0.021 | 0.027 | 0.038 | 0.033 | 0.048 |
| M4 (Monthly) | 0.101 | 0.103 | 0.103 | 0.103 | 0.100 | 0.099 | 0.098 | 0.098 | 0.153 | 0.102 | 0.100 | 0.087 | 0.095 | 0.101 | 0.097 | 0.146 |
| M4 (Weekly) | 0.037 | 0.037 | 0.040 | 0.041 | 0.040 | 0.039 | 0.036 | 0.036 | 0.078 | 0.049 | 0.047 | 0.040 | 0.039 | 0.046 | 0.051 | 0.063 |
| Pedestrian Counts | 0.187 | 0.204 | 0.237 | 0.236 | 0.227 | 0.219 | 0.210 | 0.195 | 0.262 | 0.273 | 0.259 | 0.233 | 0.257 | 0.229 | 0.261 | 0.319 |
| Rideshare | 0.140 | 0.137 | 0.140 | 0.133 | 0.136 | 0.137 | 0.137 | 0.138 | 0.158 | 0.164 | 0.159 | 0.133 | 0.135 | 0.130 | 0.134 | 0.186 |
| Taxi (30 Min.) | 0.268 | 0.274 | 0.312 | 0.313 | 0.300 | 0.284 | 0.278 | 0.267 | 0.357 | 0.513 | 0.368 | 0.334 | 0.363 | 0.395 | 0.382 | 0.471 |
| Temperature-Rain | 0.663 | 0.669 | 0.685 | 0.704 | 0.651 | 0.646 | 0.640 | 0.637 | 0.717 | 0.655 | 0.685 | 0.646 | 0.804 | 0.718 | 0.670 | 1.424 |
| Uber TLC (Daily) | 0.096 | 0.097 | 0.100 | 0.105 | 0.112 | 0.107 | 0.103 | 0.102 | 0.176 | 0.115 | 0.108 | 0.089 | 0.100 | 0.110 | 0.111 | 0.231 |
| Uber TLC (Hourly) | 0.153 | 0.153 | 0.155 | 0.161 | 0.158 | 0.157 | 0.154 | 0.160 | 0.176 | 0.176 | 0.166 | 0.153 | 0.167 | 0.176 | 0.179 | 0.299 |
| **Agg. Relative Score** | 0.564 | 0.580 | 0.603 | 0.598 | 0.592 | 0.591 | 0.573 | 0.569 | 0.937 | 0.690 | 0.669 | 0.579 | 0.601 | 0.676 | 0.734 | 1.000 |
| **Avg. Rank** | 4.867 | 6.067 | 8.467 | 8.133 | 6.800 | 5.933 | 4.867 | 4.733 | 13.333 | 12.133 | 10.333 | 5.467 | 7.267 | 10.800 | 11.400 | 15.400 |

Table 2: Raw per-dataset values of MASE for all models on the in-domain benchmark.

| | Pretrained Models (In Domain) | | | | | | | | Pretrained Models (Other) | | | | Task Specific Models | | | Local Models |
| --- | --- | --- | --- | --- | --- | --- | --- | --- | --- | --- | --- | --- | --- | --- | --- | --- |
| | Chronos (Large) | Chronos (Base) | Chronos (Small) | Chronos (Mini) | WaveToken (Mini) | WaveToken (Small) | WaveToken (Base) | WaveToken (Large) | Lag-Llama | Moirai (Base) | Moirai (Large) | TimesFM | PatchTST | DeepAR | TFT | Seasonal Naive |
| Electricity (15 Min.) | 0.391 | 0.394 | 0.418 | 0.445 | 0.443 | 0.422 | 0.410 | 0.410 | 1.169 | 0.707 | 0.625 | 0.750 | 0.450 | 0.515 | 1.108 | 0.498 |
| Electricity (Hourly) | 1.439 | 1.590 | 1.477 | 1.348 | 1.503 | 1.580 | 1.614 | 1.419 | 1.573 | 1.712 | 1.669 | 1.200 | 1.349 | 1.528 | 1.789 | 1.840 |
| Electricity (Weekly) | 1.739 | 1.801 | 1.942 | 1.954 | 1.938 | 1.890 | 1.879 | 1.864 | 2.979 | 2.858 | 2.758 | 1.773 | 1.631 | 2.517 | 2.800 | 3.037 |
| KDD Cup 2018 | 0.683 | 0.646 | 0.687 | 0.667 | 0.618 | 0.624 | 0.631 | 0.654 | 0.844 | 0.661 | 0.656 | 0.687 | 0.616 | 0.779 | 1.022 | 0.994 |
| London Smart Meters | 0.828 | 0.838 | 0.846 | 0.857 | 0.771 | 0.759 | 0.748 | 0.740 | 0.792 | 0.770 | 0.754 | 0.822 | 0.733 | 0.832 | 0.788 | 0.966 |
| M4 (Daily) | 3.144 | 3.160 | 3.148 | 3.154 | 3.140 | 3.120 | 3.145 | 3.116 | 8.038 | 3.445 | 3.377 | 3.269 | 3.450 | 3.305 | 3.292 | 3.278 |
| M4 (Hourly) | 0.682 | 0.694 | 0.721 | 0.758 | 0.722 | 0.697 | 0.667 | 0.671 | 3.807 | 1.210 | 0.951 | 0.767 | 0.967 | 1.215 | 1.833 | 1.193 |
| M4 (Monthly) | 0.960 | 0.970 | 0.982 | 0.991 | 0.992 | 0.974 | 0.956 | 0.950 | 2.090 | 1.033 | 1.005 | 0.886 | 0.962 | 1.040 | 1.009 | 1.260 |
| M4 (Weekly) | 1.998 | 2.021 | 2.113 | 2.155 | 2.137 | 2.077 | 2.005 | 1.948 | 5.658 | 2.475 | 2.419 | 2.262 | 1.996 | 2.346 | 2.745 | 2.777 |
| Pedestrian Counts | 0.272 | 0.286 | 0.304 | 0.303 | 0.309 | 0.292 | 0.292 | 0.279 | 0.342 | 0.355 | 0.330 | 0.307 | 0.827 | 0.996 | 1.067 | 1.250 |
| Rideshare | 0.865 | 0.862 | 0.854 | 0.830 | 0.840 | 0.854 | 0.856 | 0.871 | 0.891 | 0.911 | 0.900 | 0.853 | 0.827 | 0.996 | 1.067 | 1.250 |
| Taxi (30 Min.) | 0.830 | 0.849 | 0.941 | 0.944 | 0.902 | 0.861 | 0.848 | 0.823 | 1.069 | 1.374 | 1.088 | 1.054 | 1.077 | 1.158 | 1.113 | 1.160 |
| Temperature-Rain | 0.980 | 0.986 | 1.012 | 1.029 | 0.945 | 0.935 | 0.929 | 0.924 | 1.031 | 0.963 | 0.998 | 1.011 | 1.250 | 1.015 | 0.994 | 2.243 |
| Uber TLC (Daily) | 0.821 | 0.839 | 0.870 | 0.906 | 0.952 | 0.938 | 0.887 | 0.876 | 1.289 | 0.937 | 0.874 | 0.803 | 0.813 | 0.905 | 0.916 | 1.378 |
| Uber TLC (Hourly) | 0.670 | 0.673 | 0.677 | 0.689 | 0.786 | 0.777 | 0.771 | 0.779 | 0.711 | 0.728 | 0.716 | 0.677 | 0.696 | 0.703 | 0.746 | 0.931 |
| **Agg. Relative Score** | 0.695 | 0.706 | 0.727 | 0.732 | 0.729 | 0.718 | 0.708 | 0.698 | 1.141 | 0.856 | 0.806 | 0.745 | 0.740 | 0.821 | 0.939 | 1.000 |
| **Avg. Rank** | 4.333 | 5.733 | 7.400 | 8.067 | 7.333 | 6.400 | 5.133 | 4.133 | 13.200 | 11.867 | 10.133 | 6.933 | 6.333 | 11.467 | 12.867 | 14.667 |

Table 3: Raw per-dataset values of VRSE for all models on the in-domain benchmark.

| | Pretrained Models (In Domain) | | | | | | | | Pretrained Models (Other) | | | | Task Specific Models | | | Local Models |
| --- | --- | --- | --- | --- | --- | --- | --- | --- | --- | --- | --- | --- | --- | --- | --- | --- |
| | Chronos (Large) | Chronos (Base) | Chronos (Small) | Chronos (Mini) | WaveToken (Mini) | WaveToken (Small) | WaveToken (Base) | WaveToken (Large) | Lag-Llama | Moirai (Base) | Moirai (Large) | TimesFM | PatchTST | DeepAR | TFT | Seasonal Naive |
| Electricity (15 Min.) | 0.022 | 0.023 | 0.021 | 0.024 | 0.027 | 0.026 | 0.024 | 0.027 | 1.354 | 0.018 | 0.024 | 0.033 | 0.022 | 0.030 | 0.109 | 0.032 |
| Electricity (Hourly) | 0.012 | 0.013 | 0.012 | 0.009 | 0.013 | 0.010 | 0.012 | 0.013 | 0.008 | 0.014 | 0.01 | 0.01 | 0.007 | 0.008 | 0.010 | 0.011 |
| Electricity (Weekly) | 0.015 | 0.014 | 0.022 | 0.021 | 0.021 | 0.019 | 0.016 | 0.014 | 0.099 | 0.093 | 0.179 | 0.011 | 0.026 | 0.070 | 0.057 | 0.159 |
| KDD Cup 2018 | 0.099 | 0.101 | 0.108 | 0.115 | 0.113 | 0.123 | 0.123 | 0.131 | 0.157 | 0.129 | 0.119 | 0.151 | 0.241 | 0.309 | 0.261 | 0.177 |
| London Smart Meters | 0.323 | 0.337 | 0.341 | 0.352 | 0.213 | 0.213 | 0.208 | 0.205 | 0.252 | 0.208 | 0.188 | 0.274 | 0.117 | 0.131 | 0.996 | 0.284 |
| M4 (Daily) | 0.010 | 0.007 | 0.005 | 0.007 | 0.004 | 0.004 | 0.004 | 0.004 | 0.022 | 0.005 | 0.004 | 0.004 | 0.005 | 0.006 | 0.007 | 0.007 |
| M4 (Hourly) | 0.000 | 0.000 | 0.001 | 0.000 | 0.000 | 0.001 | 0.001 | 0.001 | 0.003 | 0.000 | 0.000 | 0.000 | 0.001 | 0.001 | 0.001 | 0.002 |
| M4 (Monthly) | 0.047 | 0.047 | 0.048 | 0.047 | 0.045 | 0.044 | 0.044 | 0.043 | 0.080 | 0.042 | 0.042 | 0.040 | 0.043 | 0.041 | 0.044 | 0.048 |
| M4 (Weekly) | 0.003 | 0.003 | 0.004 | 0.004 | 0.004 | 0.004 | 0.003 | 0.003 | 0.020 | 0.006 | 0.005 | 0.004 | 0.004 | 0.005 | 0.008 | 0.008 |
| Pedestrian Counts | 0.083 | 0.104 | 0.110 | 0.108 | 0.111 | 0.109 | 0.105 | 0.099 | 0.121 | 0.126 | 0.133 | 0.111 | 0.128 | 0.088 | 0.109 | 0.118 |
| Rideshare | 0.043 | 0.041 | 0.043 | 0.041 | 0.043 | 0.044 | 0.044 | 0.043 | 0.052 | 0.066 | 0.064 | 0.049 | 0.041 | 0.031 | 0.049 | 0.032 |
| Taxi (30 Min.) | 0.097 | 0.099 | 0.136 | 0.126 | 0.118 | 0.104 | 0.096 | 0.087 | 0.169 | 0.329 | 0.156 | 0.172 | 0.214 | 0.208 | 0.222 | 0.108 |
| Temperature-Rain | 0.596 | 0.633 | 0.677 | 0.712 | 0.512 | 0.515 | 0.505 | 0.493 | 0.423 | 0.419 | 0.409 | 0.528 | 0.854 | 0.527 | 0.542 | 1.715 |
| Uber TLC (Daily) | 0.024 | 0.023 | 0.023 | 0.024 | 0.028 | 0.023 | 0.024 | 0.024 | 0.067 | 0.031 | 0.026 | 0.024 | 0.023 | 0.024 | 0.024 | 0.115 |
| Uber TLC (Hourly) | 0.020 | 0.019 | 0.020 | 0.024 | 0.022 | 0.021 | 0.020 | 0.024 | 0.047 | 0.029 | 0.023 | 0.030 | 0.030 | 0.030 | 0.028 | 0.058 |
| **Agg. Relative Score** | 0.550 | 0.547 | 0.582 | 0.591 | 0.555 | 0.567 | 0.540 | 0.540 | 1.350 | 0.684 | 0.664 | 0.548 | 0.608 | 0.681 | 0.921 | 1.000 |
| **Avg. Rank** | 6.733 | 6.733 | 8.267 | 8.400 | 8.000 | 6.667 | 6.400 | 6.467 | 12.733 | 9.533 | 8.467 | 6.800 | 8.533 | 8.867 | 11.667 | 11.733 |

# E ADDITIONAL RESULTS: LONG-HORIZON FORECASTING

Figure 12 shows results on a long-horizon benchmark constructed by increasing the forecast length $H$ of each dataset in Benchmark II (Zero-Shot) by a factor of 2 and 3, implying maximum horizons of 112 and 168 (for NN5 Daily). Benchmark I (In-Domain) was not used as increasing the forecast length over the prescribed ones (see Table 9) would imply mixing the test and training portions of the datasets. The datasets in Benchmark II without sufficient history in all time series to allow for longer horizons have been skipped. WaveToken-Base outperforms other foundation models across all three metrics in the $H \times 2$ setting. When $H \times 3$, TimesFM performs better with respect to WQL only. All results for Chronos and WaveToken are averaged across three different seeds.

# F ADDITIONAL RESULTS: QUALITATIVE ANALYSIS

Figures 13, 14 and 15 show all the cross-attention maps for the 12 decoder layers of Chronos-Base (left) and WaveToken-Base (right) when forecasting the spiky data of the second row in Figure 1.

## Table 4: Raw per-dataset values of WQL for all models on the zero-shot benchmark.

|  | Pretrained Models (Zero Shot) | | | | | | | | Pretrained Models (Other) | | | | Task Specific Models | | | Local Models |
|---|---|---|---|---|---|---|---|---|---|---|---|---|---|---|---|---|
|  | Chronos (Large) | Chronos (Base) | Chronos (Small) | Chronos (Mini) | WaveToken (Mini) | WaveToken (Small) | WaveToken (Base) | WaveToken (Large) | Lag-Llama | Moirai (Base) | Moirai (Large) | TimesFM | PatchTST | DeepAR | TFT | Seasonal Naive |
| Australian Electricity | 0.067 | 0.075 | 0.074 | 0.063 | 0.094 | 0.065 | 0.075 | 0.071 | 0.097 | 0.055 | 0.046 | 0.089 | 0.037 | 0.087 | 0.036 | 0.084 |
| Car Parts | 1.060 | 1.057 | 1.029 | 1.024 | 0.973 | 0.986 | 0.994 | 1.001 | 1.011 | 1.654 | 1.621 | 0.998 | 0.967 | 0.871 |  | 1.600 |
| CIF 2016 | 0.014 | 0.013 | 0.015 | 0.013 | 0.014 | 0.012 | 0.014 | 0.012 | 0.041 | 0.011 | 0.031 | 0.020 | 0.140 | 0.136 | 0.011 | 0.015 |
| Covid Deaths | 0.045 | 0.048 | 0.059 | 0.084 | 0.057 | 0.050 | 0.048 | 0.049 | 0.276 | 0.038 | 0.035 | 0.204 | 0.065 | 0.108 | 0.034 | 0.133 |
| Dominick | 0.332 | 0.333 | 0.338 | 0.346 | 0.352 | 0.348 | 0.343 | 0.342 | 0.443 | 0.361 | 0.346 | 0.426 | 0.345 | 0.364 | 0.320 | 0.453 |
| ERCOT Load | 0.019 | 0.016 | 0.018 | 0.018 | 0.017 | 0.015 | 0.015 | 0.019 | 0.033 | 0.019 | 0.021 | 0.021 | 0.017 | 0.032 | 0.023 | 0.037 |
| ETT (15 Min.) | 0.068 | 0.069 | 0.064 | 0.072 | 0.068 | 0.071 | 0.063 | 0.061 | 0.080 | 0.075 | 0.070 | 0.084 | 0.054 | 0.069 | 0.075 | 0.141 |
| ETT (Hourly) | 0.073 | 0.081 | 0.080 | 0.085 | 0.080 | 0.072 | 0.073 | 0.074 | 0.106 | 0.095 | 0.084 | 0.092 | 0.071 | 0.081 | 0.082 | 0.122 |
| Exchange Rate | 0.013 | 0.014 | 0.013 | 0.012 | 0.015 | 0.014 | 0.014 | 0.013 | 0.011 | 0.010 | 0.012 | 0.013 | 0.010 | 0.009 | 0.011 | 0.013 |
| FRED-MD | 0.020 | 0.022 | 0.017 | 0.017 | 0.026 | 0.022 | 0.027 | 0.026 | 0.389 | 0.047 | 0.048 | 0.035 | 0.042 | 0.043 | 0.112 | 0.122 |
| Hospital | 0.056 | 0.056 | 0.057 | 0.058 | 0.060 | 0.059 | 0.059 | 0.055 | 0.093 | 0.060 | 0.057 | 0.051 | 0.070 | 0.056 | 0.053 | 0.073 |
| M1 (Monthly) | 0.130 | 0.128 | 0.139 | 0.138 | 0.140 | 0.139 | 0.134 | 0.128 | 0.196 | 0.155 | 0.151 | 0.123 | 0.165 | 0.150 | 0.175 | 0.191 |
| M1 (Quarterly) | 0.107 | 0.105 | 0.103 | 0.103 | 0.111 | 0.110 | 0.111 | 0.119 | 0.141 | 0.108 | 0.107 | 0.087 | 0.078 | 0.089 | 0.122 | 0.150 |
| M1 (Yearly) | 0.183 | 0.181 | 0.172 | 0.179 | 0.177 | 0.173 | 0.167 | 0.166 | 0.293 | 0.195 | 0.199 | 0.163 | 0.165 | 0.139 | 0.124 | 0.209 |
| M3 (Monthly) | 0.096 | 0.097 | 0.100 | 0.099 | 0.099 | 0.098 | 0.097 | 0.095 | 0.155 | 0.102 | 0.101 | 0.093 | 0.113 | 0.099 | 0.096 | 0.149 |
| M3 (Quarterly) | 0.074 | 0.076 | 0.079 | 0.081 | 0.079 | 0.079 | 0.077 | 0.076 | 0.134 | 0.080 | 0.085 | 0.072 | 0.074 | 0.073 | 0.071 | 0.101 |
| M3 (Yearly) | 0.151 | 0.153 | 0.155 | 0.159 | 0.140 | 0.149 | 0.151 | 0.143 | 0.192 | 0.166 | 0.170 | 0.123 | 0.133 | 0.122 | 0.130 | 0.167 |
| M4 (Quarterly) | 0.082 | 0.083 | 0.084 | 0.086 | 0.081 | 0.081 | 0.080 | 0.079 | 0.132 | 0.081 | 0.080 | 0.074 | 0.074 | 0.080 | 0.080 | 0.119 |
| M4 (Yearly) | 0.134 | 0.137 | 0.136 | 0.140 | 0.134 | 0.136 | 0.134 | 0.130 | 0.178 | 0.121 | 0.138 | 0.117 | 0.106 | 0.111 | 0.110 | 0.161 |
| M5 | 0.587 | 0.586 | 0.590 | 0.595 | 0.602 | 0.597 | 0.593 | 0.590 | 0.635 | 0.692 | 0.584 | 0.559 | 0.597 | 0.657 | 0.560 | 1.024 |
| NN5 (Daily) | 0.156 | 0.161 | 0.169 | 0.173 | 0.190 | 0.176 | 0.162 | 0.161 | 0.261 | 0.181 | 0.162 | 0.160 | 0.149 | 0.155 | 0.145 | 0.425 |
| NN5 (Weekly) | 0.091 | 0.091 | 0.090 | 0.091 | 0.094 | 0.092 | 0.092 | 0.090 | 0.111 | 0.092 | 0.092 | 0.086 | 0.081 | 0.087 | 0.086 | 0.123 |
| Tourism (Monthly) | 0.100 | 0.103 | 0.113 | 0.109 | 0.117 | 0.103 | 0.103 | 0.096 | 0.213 | 0.123 | 0.113 | 0.088 | 0.092 | 0.092 | 0.096 | 0.104 |
| Tourism (Quarterly) | 0.061 | 0.069 | 0.069 | 0.074 | 0.071 | 0.070 | 0.066 | 0.063 | 0.202 | 0.100 | 0.085 | 0.074 | 0.074 | 0.072 | 0.074 | 0.119 |
| Tourism (Yearly) | 0.183 | 0.207 | 0.200 | 0.218 | 0.184 | 0.197 | 0.193 | 0.192 | 0.238 | 0.167 | 0.163 | 0.148 | 0.136 | 0.127 | 0.102 | 0.209 |
| Traffic | 0.256 | 0.264 | 0.263 | 0.264 | 0.240 | 0.250 | 0.260 | 0.250 | 0.256 | 0.225 | 0.231 | 0.184 | 0.246 | 0.233 | 0.264 | 0.362 |
| Weather | 0.139 | 0.140 | 0.143 | 0.150 | 0.142 | 0.140 | 0.139 | 0.137 | 0.164 | 0.135 | 0.132 | 0.150 | 0.143 | 0.147 | 0.151 | 0.217 |
| **Agg. Relative Score** | 0.645 | 0.662 | 0.667 | 0.678 | 0.683 | 0.655 | 0.655 | 0.644 | 1.097 | 0.698 | 0.707 | 0.692 | 0.684 | 0.733 | 0.639 | 1.000 |
| **Avg. Rank** | 6.630 | 7.963 | 8.556 | 9.741 | 9.333 | 8.000 | 7.074 | 5.630 | 14.481 | 9.667 | 9.296 | 6.259 | 6.037 | 7.000 | 5.778 | 14.556 |

## Table 5: Raw per-dataset values of MASE for all models on the zero-shot benchmark.

|  | Pretrained Models (Zero Shot) | | | | | | | | Pretrained Models (Other) | | | | Task Specific Models | | | Local Models |
|---|---|---|---|---|---|---|---|---|---|---|---|---|---|---|---|---|
|  | Chronos (Large) | Chronos (Base) | Chronos (Small) | Chronos (Mini) | WaveToken (Mini) | WaveToken (Small) | WaveToken (Base) | WaveToken (Large) | Lag-Llama | Moirai (Base) | Moirai (Large) | TimesFM | PatchTST | DeepAR | TFT | Seasonal Naive |
| Australian Electricity | 1.333 | 1.319 | 1.399 | 1.114 | 1.657 | 1.287 | 1.404 | 1.310 | 1.635 | 1.250 | 0.995 | 1.631 | 0.871 | 1.473 | 0.810 | 1.253 |
| Car Parts | 0.906 | 0.899 | 0.887 | 0.891 | 0.847 | 0.860 | 0.865 | 0.873 | 0.816 | 1.734 | 1.540 | 0.893 | 0.803 | 0.798 | 0.799 | 1.201 |
| CIF 2016 | 0.986 | 0.981 | 0.989 | 1.051 | 1.066 | 1.012 | 0.997 | 0.977 | 2.235 | 1.208 | 1.138 | 0.925 | 1.537 | 1.363 | 1.553 | 1.289 |
| Covid Deaths | 42.550 | 42.687 | 42.670 | 43.621 | 38.051 | 37.939 | 37.670 | 36.969 | 78.456 | 33.036 | 33.063 | 55.627 | 36.465 | 38.203 | 30.635 | 46.912 |
| Dominick | 0.818 | 0.816 | 0.819 | 0.833 | 0.825 | 0.823 | 0.821 | 0.822 | 1.250 | 0.880 | 0.845 | 1.220 | 0.867 | 0.851 | 0.800 | 0.871 |
| ERCOT Load | 0.617 | 0.550 | 0.573 | 0.588 | 0.565 | 0.510 | 0.519 | 0.627 | 0.834 | 0.590 | 0.660 | 0.590 | 0.553 | 1.197 | 0.690 | 0.761 |
| ETT (15 Min.) | 0.741 | 0.739 | 0.710 | 0.792 | 0.729 | 0.710 | 0.674 | 0.642 | 0.967 | 0.968 | 0.765 | 1.037 | 0.652 | 0.874 | 0.962 | 1.169 |
| ETT (Hourly) | 0.735 | 0.789 | 0.780 | 0.797 | 0.795 | 0.737 | 0.723 | 0.726 | 1.002 | 0.895 | 0.839 | 0.890 | 0.729 | 0.814 | 0.875 | 0.932 |
| Exchange Rate | 2.375 | 2.433 | 2.252 | 2.030 | 2.306 | 2.268 | 2.224 | 2.348 | 3.087 | 1.536 | 1.923 | 3.310 | 1.540 | 1.615 | 2.361 | 1.740 |
| FRED-MD | 0.500 | 0.486 | 0.496 | 0.483 | 0.510 | 0.503 | 0.501 | 0.500 | 2.283 | 0.609 | 0.598 | 0.484 | 0.745 | 0.621 | 0.929 | 1.101 |
| Hospital | 0.810 | 0.810 | 0.815 | 0.817 | 0.713 | 0.709 | 0.697 | 0.697 | 0.939 | 0.821 | 0.824 | 0.759 | 0.859 | 0.804 | 0.799 | 0.921 |
| M1 (Monthly) | 1.090 | 1.117 | 1.169 | 1.174 | 1.205 | 1.158 | 1.104 | 1.079 | 1.875 | 1.271 | 1.241 | 1.027 | 1.208 | 1.122 | 1.326 | 1.314 |
| M1 (Quarterly) | 1.713 | 1.739 | 1.764 | 1.785 | 1.782 | 1.779 | 1.758 | 1.787 | 1.877 | 1.877 | 1.829 | 1.632 | 1.920 | 1.741 | 2.144 | 2.078 |
| M1 (Yearly) | 4.301 | 4.624 | 4.659 | 4.958 | 4.737 | 4.895 | 4.751 | 4.449 | 7.149 | 4.629 | 4.707 | 4.004 | 4.042 | 3.685 | 4.316 | 4.894 |
| M3 (Monthly) | 0.857 | 0.868 | 0.885 | 0.900 | 0.912 | 0.888 | 0.877 | 0.858 | 1.846 | 0.947 | 0.924 | 0.870 | 1.225 | 0.943 | 0.916 | 1.146 |
| M3 (Quarterly) | 1.181 | 1.199 | 1.256 | 1.289 | 1.272 | 1.247 | 1.223 | 1.216 | 2.886 | 1.433 | 1.439 | 1.150 | 1.264 | 1.209 | 1.160 | 1.439 |
| M3 (Yearly) | 3.106 | 3.209 | 3.276 | 3.385 | 3.009 | 3.142 | 3.185 | 2.992 | 5.114 | 3.654 | 3.823 | 2.697 | 2.949 | 2.827 | 2.860 | 3.172 |
| M4 (Quarterly) | 1.216 | 1.231 | 1.246 | 1.271 | 1.243 | 1.226 | 1.212 | 1.210 | 2.663 | 1.285 | 1.259 | 1.160 | 1.210 | 1.254 | 1.248 | 1.602 |
| M4 (Yearly) | 3.606 | 3.678 | 3.651 | 3.743 | 3.688 | 3.731 | 3.652 | 3.550 | 5.866 | 3.601 | 4.174 | 3.339 | 3.072 | 3.178 | 3.119 | 3.974 |
| M5 | 0.944 | 0.939 | 0.940 | 0.944 | 0.944 | 0.945 | 0.941 | 0.940 | 0.965 | 1.440 | 0.930 | 0.912 | 0.919 | 0.986 | 0.909 | 1.399 |
| NN5 (Daily) | 0.573 | 0.585 | 0.615 | 0.642 | 0.732 | 0.678 | 0.621 | 0.615 | 0.992 | 0.700 | 0.626 | 0.629 | 0.575 | 0.585 | 0.556 | 1.292 |
| NN5 (Weekly) | 0.940 | 0.938 | 0.944 | 0.947 | 0.966 | 0.951 | 0.948 | 0.937 | 1.141 | 0.991 | 0.994 | 0.949 | 0.877 | 0.920 | 0.896 | 1.063 |
| Tourism (Monthly) | 1.761 | 1.828 | 1.900 | 1.950 | 1.997 | 1.821 | 1.695 | 1.633 | 3.030 | 2.040 | 1.911 | 1.541 | 1.572 | 1.529 | 1.686 | 1.631 |
| Tourism (Quarterly) | 1.677 | 1.717 | 1.730 | 1.829 | 1.836 | 1.804 | 1.762 | 1.725 | 3.695 | 2.712 | 2.306 | 1.731 | 1.723 | 1.586 | 1.729 | 1.699 |
| Tourism (Yearly) | 3.755 | 3.900 | 3.901 | 4.048 | 3.635 | 3.852 | 3.697 | 3.689 | 5.866 | 3.755 | 3.060 | 3.284 | 3.234 | 3.138 | 3.702 | 3.552 |
| Traffic | 0.804 | 0.828 | 0.837 | 0.850 | 0.785 | 0.817 | 0.843 | 0.817 | 0.829 | 0.725 | 0.759 | 0.638 | 0.790 | 0.737 | 0.880 | 1.077 |
| Weather | 0.822 | 0.824 | 0.836 | 0.853 | 0.846 | 0.834 | 0.830 | 0.822 | 1.001 | 0.831 | 0.808 | 0.912 | 0.860 | 0.911 | 0.913 | 1.004 |
| **Agg. Relative Score** | 0.823 | 0.832 | 0.841 | 0.850 | 0.848 | 0.828 | 0.817 | 0.810 | 1.291 | 0.908 | 0.875 | 0.846 | 0.810 | 0.843 | 0.847 | 1.000 |
| **Avg. Rank** | 6.333 | 6.926 | 8.148 | 10.111 | 9.111 | 8.037 | 6.741 | 5.481 | 14.815 | 10.556 | 9.815 | 6.741 | 6.148 | 7.296 | 7.333 | 12.407 |

## Table 6: Raw per-dataset values of VRSE for all models on the zero-shot benchmark.

|  | Pretrained Models (Zero Shot) | | | | | | | | Pretrained Models (Other) | | | | Task Specific Models | | | Local Models |
|---|---|---|---|---|---|---|---|---|---|---|---|---|---|---|---|---|
|  | Chronos (Large) | Chronos (Base) | Chronos (Small) | Chronos (Mini) | WaveToken (Mini) | WaveToken (Small) | WaveToken (Base) | WaveToken (Large) | Lag-Llama | Moirai (Base) | Moirai (Large) | TimesFM | PatchTST | DeepAR | TFT | Seasonal Naive |
| Australian Electricity | 0.008 | 0.008 | 0.008 | 0.010 | 0.012 | 0.005 | 0.009 | 0.010 | 0.006 | 0.004 | 0.005 | 0.013 | 0.002 | 0.012 | 0.001 | 0.008 |
| Car Parts | 0.906 | 0.922 | 0.867 | 0.843 | 0.863 | 0.854 | 0.845 | 0.848 | 0.933 | 0.967 | 0.946 | 0.893 | 0.976 | 0.908 | 0.948 | 0.827 |
| CIF 2016 | 0.000 | 0.000 | 0.000 | 0.000 | 0.000 | 0.000 | 0.000 | 0.000 | 0.005 | 0.000 | 0.000 | 0.001 | 0.035 | 0.030 | 0.000 | 0.000 |
| Covid Deaths | 0.003 | 0.004 | 0.006 | 0.012 | 0.006 | 0.005 | 0.003 | 0.003 | 0.079 | 0.003 | 0.009 | 0.099 | 0.011 | 0.045 | 0.002 | 0.026 |
| Dominick | 0.273 | 0.279 | 0.290 | 0.308 | 0.242 | 0.243 | 0.241 | 0.245 | 0.283 | 0.244 | 0.242 | 0.204 | 0.279 | 0.271 | 0.311 | 0.401 |
| ERCOT Load | 0.001 | 0.001 | 0.001 | 0.001 | 0.001 | 0.000 | 0.001 | 0.001 | 0.004 | 0.001 | 0.001 | 0.001 | 0.001 | 0.003 | 0.001 | 0.002 |
| ETT (15 Min.) | 0.006 | 0.006 | 0.006 | 0.008 | 0.007 | 0.007 | 0.006 | 0.005 | 0.034 | 0.007 | 0.006 | 0.010 | 0.004 | 0.006 | 0.006 | 0.019 |
| ETT (Hourly) | 0.007 | 0.007 | 0.008 | 0.009 | 0.007 | 0.006 | 0.006 | 0.007 | 0.020 | 0.012 | 0.009 | 0.014 | 0.007 | 0.009 | 0.010 | 0.010 |
| Exchange Rate | 0.000 | 0.000 | 0.000 | 0.000 | 0.000 | 0.000 | 0.000 | 0.000 | 0.000 | 0.000 | 0.000 | 0.000 | 0.000 | 0.000 | 0.000 | 0.000 |
| FRED-MD | 0.005 | 0.001 | 0.000 | 0.000 | 0.004 | 0.002 | 0.006 | 0.004 | 0.577 | 0.013 | 0.003 | 0.004 | 0.010 | 0.009 | 0.009 | 0.051 |
| Hospital | 0.003 | 0.003 | 0.004 | 0.004 | 0.004 | 0.004 | 0.003 | 0.003 | 0.007 | 0.003 | 0.004 | 0.003 | 0.006 | 0.009 | 0.010 | 0.005 |
| M1 (Monthly) | 0.059 | 0.060 | 0.057 | 0.061 | 0.078 | 0.082 | 0.071 | 0.060 | 0.177 | 0.100 | 0.078 | 0.106 | 0.174 | 0.156 | 0.232 | 0.094 |
| M1 (Quarterly) | 0.020 | 0.020 | 0.020 | 0.015 | 0.021 | 0.020 | 0.022 | 0.027 | 0.031 | 0.019 | 0.018 | 0.016 | 0.008 | 0.008 | 0.028 | 0.029 |
| M1 (Yearly) | 0.097 | 0.094 | 0.079 | 0.087 | 0.084 | 0.082 | 0.078 | 0.072 | 0.193 | 0.094 | 0.081 | 0.077 | 0.068 | 0.039 | 0.037 | 0.092 |
| M3 (Monthly) | 0.032 | 0.033 | 0.037 | 0.031 | 0.030 | 0.030 | 0.029 | 0.028 | 0.064 | 0.030 | 0.031 | 0.031 | 0.039 | 0.034 | 0.032 | 0.036 |
| M3 (Quarterly) | 0.024 | 0.026 | 0.028 | 0.029 | 0.026 | 0.025 | 0.025 | 0.024 | 0.059 | 0.024 | 0.024 | 0.023 | 0.021 | 0.021 | 0.021 | 0.026 |
| M3 (Yearly) | 0.095 | 0.079 | 0.079 | 0.086 | 0.062 | 0.062 | 0.086 | 0.067 | 0.064 | 0.051 | 0.055 | 0.060 | 0.066 | 0.061 | 0.060 | 0.037 |
| M4 (Yearly) | 0.060 | 0.060 | 0.060 | 0.062 | 0.060 | 0.061 | 0.060 | 0.058 | 0.193 | 0.050 | 0.057 | 0.056 | 0.051 | 0.048 | 0.052 | 0.065 |
| M5 | 0.279 | 0.275 | 0.286 | 0.285 | 0.319 | 0.296 | 0.292 | 0.281 | 0.308 | 0.258 | 0.280 | 0.324 | 0.375 | 0.338 | 0.318 | 0.618 |
| NN5 (Daily) | 0.047 | 0.046 | 0.046 | 0.044 | 0.041 | 0.041 | 0.040 | 0.040 | 0.108 | 0.046 | 0.038 | 0.053 | 0.043 | 0.035 | 0.042 | 0.275 |
| NN5 (Weekly) | 0.016 | 0.016 | 0.016 | 0.016 | 0.017 | 0.021 | 0.016 | 0.016 | 0.021 | 0.016 | 0.014 | 0.009 | 0.013 | 0.015 | 0.019 | 0.010 |
| Tourism (Monthly) | 0.019 | 0.021 | 0.027 | 0.015 | 0.021 | 0.019 | 0.020 | 0.026 | 0.045 | 0.016 | 0.014 | 0.019 | 0.005 | 0.004 | 0.019 | 0.024 |
| Tourism (Quarterly) | 0.004 | 0.004 | 0.004 | 0.005 | 0.004 | 0.004 | 0.003 | 0.002 | 0.045 | 0.005 | 0.002 | 0.002 | 0.005 | 0.005 | 0.009 | 0.024 |
| Tourism (Yearly) | 0.033 | 0.037 | 0.032 | 0.036 | 0.030 | 0.035 | 0.038 | 0.036 | 0.125 | 0.038 | 0.02 | 0.021 | 0.014 | 0.018 | 0.007 | 0.034 |
| Traffic | 0.243 | 0.244 | 0.243 | 0.239 | 0.227 | 0.235 | 0.241 | 0.236 | 0.224 | 0.223 | 0.221 | 0.150 | 0.227 | 0.223 | 0.247 | 0.279 |
| Weather | 0.070 | 0.070 | 0.072 | 0.073 | 0.068 | 0.068 | 0.068 | 0.067 | 0.073 | 0.065 | 0.065 | 0.078 | 0.076 | 0.071 | 0.079 | 0.079 |
| **Agg. Relative Score** | 0.623 | 0.629 | 0.608 | 0.624 | 0.632 | 0.585 | 0.610 | 0.584 | 1.843 | 0.604 | 0.590 | 0.773 | 0.732 | 0.854 | 0.667 | 1.000 |
| **Avg. Rank** | 8.407 | 9.296 | 9.333 | 9.444 | 8.370 | 7.333 | 7.111 | 6.444 | 13.889 | 6.481 | 6.47 | 8.630 | 7.667 | 7.704 | 9.111 | 11.667 |

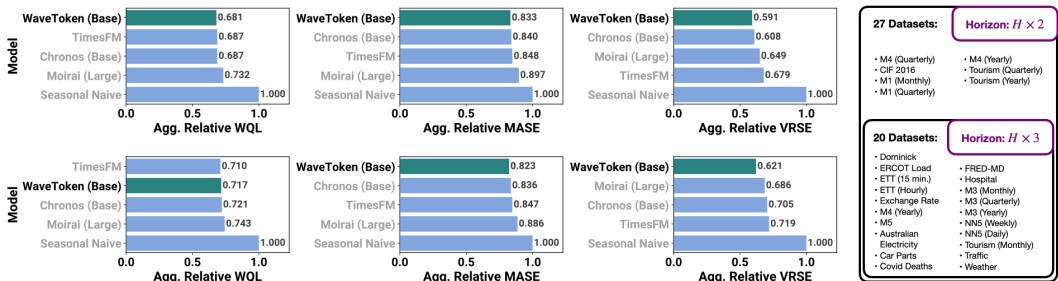

Figure 12: **Long-horizon benchmark constructed by increasing the forecast length of each dataset in Benchmark II (Zero-Shot) by a factor of 2 and 3.** *(Top row)* Results with horizon multiplied by 2: WaveToken-Base outperforms other foundation models across all three metrics. *(Bottom row)* Results with horizon multiplied by 3: WaveToken-Base outperforms other foundation models with respect to MASE and VRSE. TimesFM performs better with respect to WQL. *(Right)* List of all datasets included in the two long-horizons benchmarks. All results for Chronos and WaveToken are averaged across three different seeds.

## G    DATASETS AND BASELINES

In this section we report details on all the datasets (Tables 7 and 9) and baselines (Table 8) used for the experiments of Section 4. For the SeasonalNaive baseline, we relied on the implementation available in the StatsForecast library (Garza et al., 2022). For the task-specific deep learning models, we used their implementations available in the GluonTS library (Alexandrov et al., 2020). Finally, we used the corresponding reference implementations for Lag-llama[5] (Rasul et al., 2023), Moirai[6] (Woo et al., 2024), TimesFM[7] (Das et al., 2023) and Chronos[8] (Ansari et al., 2024).

Table 7: Breakdown of the datasets and baselines used for training and evaluation.

| Data Subset | # Datasets | # Series | Usage | Evaluation Baselines |
|---|---|---|---|---|
| Pretraining-only | 13 | 795,936 | pretraining | - |
| Benchmark I | 15 | 97,272 | pretraining and in-domain evaluation | SeasonalNaive, DeepAR, TFT, PatchTST Lag-Llama, Moirai-1.0-R (Base & Large), Chronos (Mini-Large), TimesFM |
| Benchmark II | 27 | 190,674 | zero-shot evaluation | All of the above |

Table 8: Baseline models and hyper-parameter choices. Hyper-parameters not specified are set to defaults in their respective implementations. $C$ stands for context length, $d_h$ for hidden layer dimension, $n_L$ for number of layers, and $n_H$ for number of heads.

| Model | Model Type | Implementation | Probabilistic | Hyperparameters |
|---|---|---|---|---|
| SeasonalNaive | Local | StatsForecast | Yes | N/A |
| DeepAR | Task-specific | GluonTS | Yes | $d_h = 40, n_L = 2$ |
| TFT | Task-specific | GluonTS | Yes | $d_h = 32, n_H = 4$ |
| PatchTST | Task-specific | GluonTS | Yes | Patch length: 16, Stride: 8, $d_h = 32, n_L = 2, n_H = 4$ |
| Lag-Llama | Pretrained | Reference | Yes | $C = 32$ |
| Moirai-1.0-R | Pretrained | Reference | Yes | $C = 1024$, Patch length: selected by dataset-specific validation |
| TimesFM | Pretrained | Reference | Yes | All hyperparameters set to defaults from the released models |
| Chronos | Pretrained | Reference | Yes | All hyperparameters set to defaults from the released models |
| WaveToken | Pretrained | Reference | Yes | All hyperparameters set to defaults as in Chronos |

---

[5] https://github.com/time-series-foundation-models/lag-llama

[6] https://github.com/SalesforceAIResearch/uni2ts

[7] https://github.com/google-research/timesfm

[8] https://github.com/amazon-science/chronos-forecasting

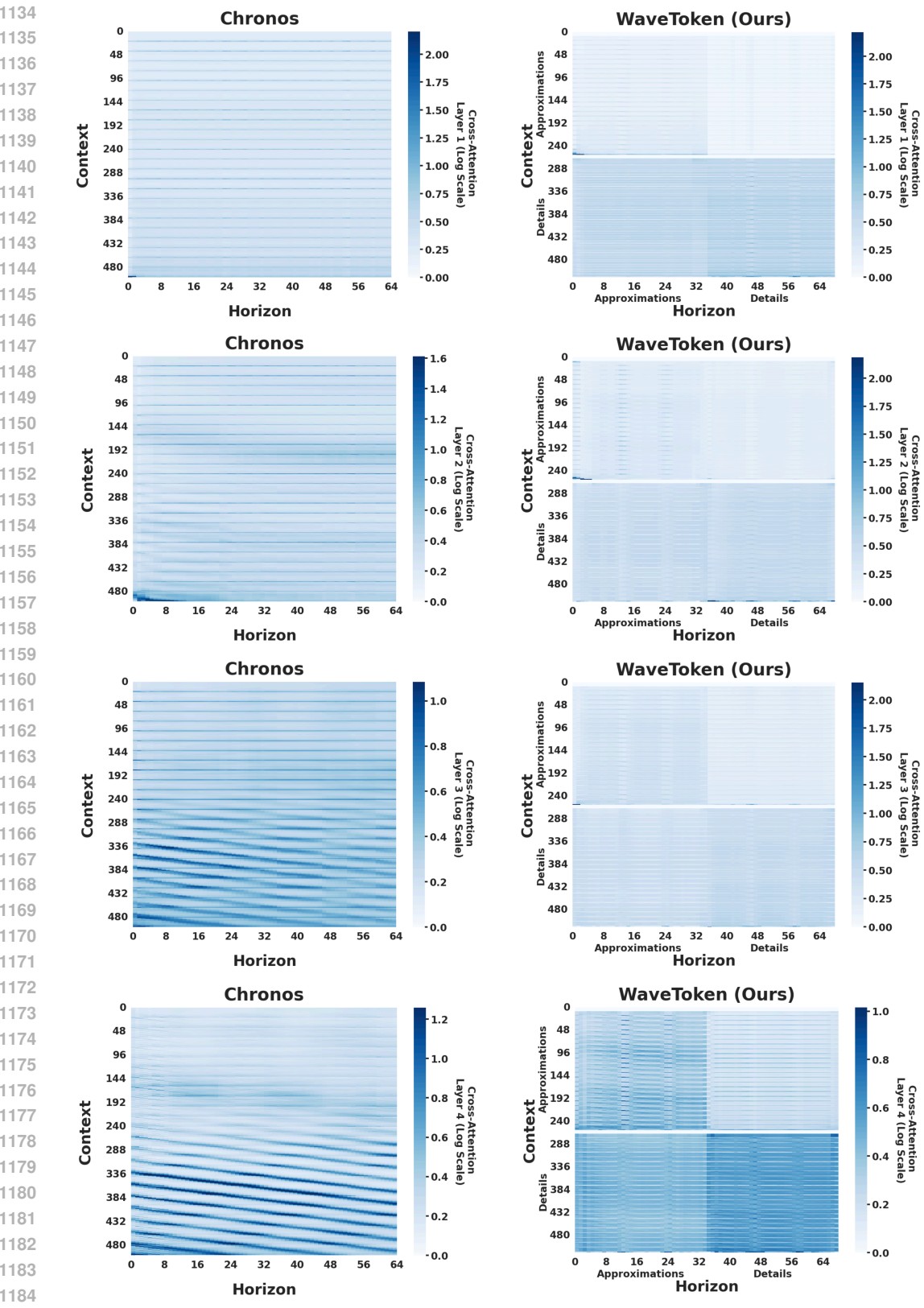

Figure 13: Cross-attention maps for the first to fourth (from top) decoder layers of Chronos-Base (left) and WaveToken-Base (right) when forecasting the spiky data of the second row in Figure 1.

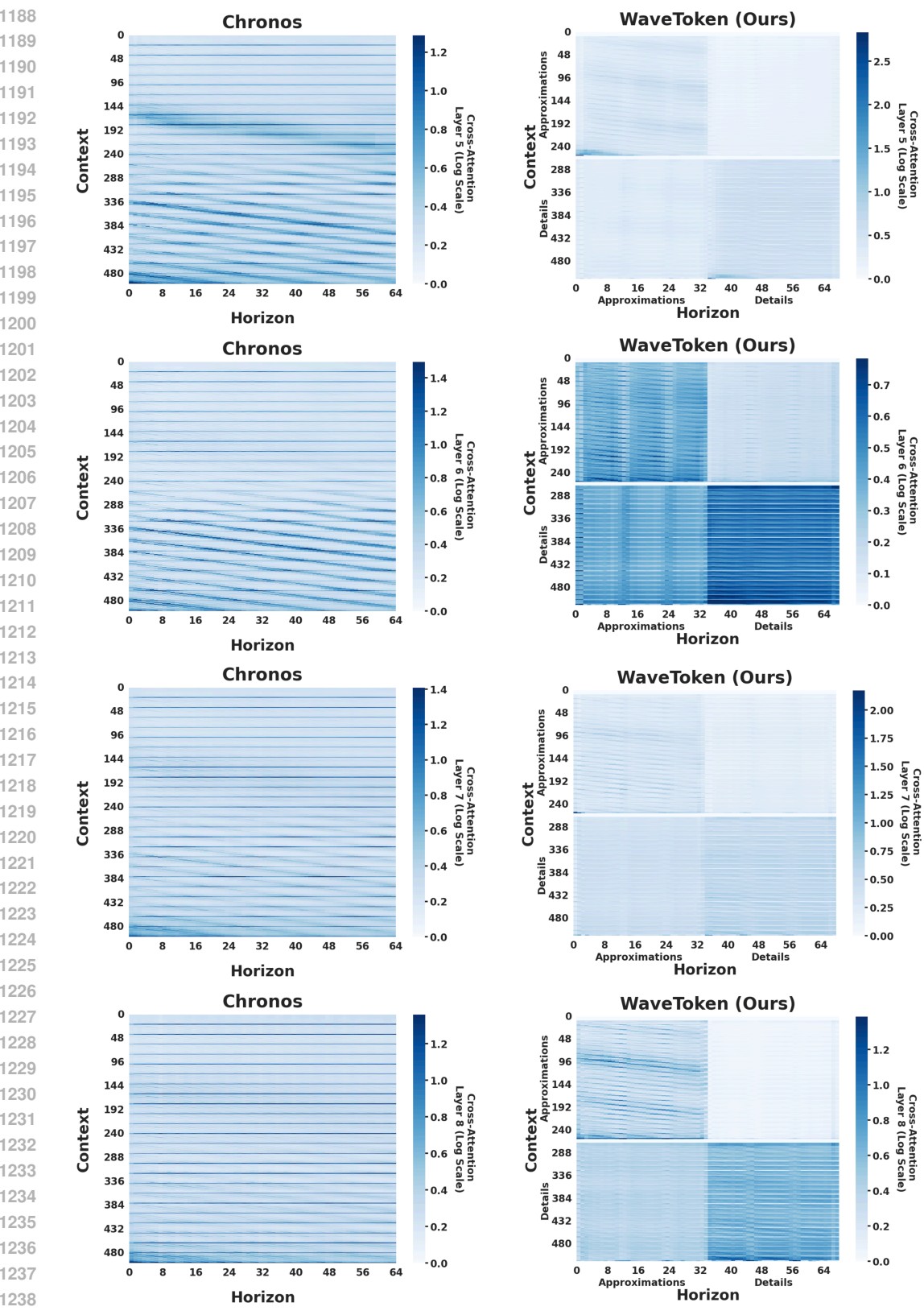

Figure 14: Cross-attention maps for the fifth to eighth (from top) decoder layers of Chronos-Base (left) and WaveToken-Base (right) when forecasting the spiky data of the second row in Figure 1.

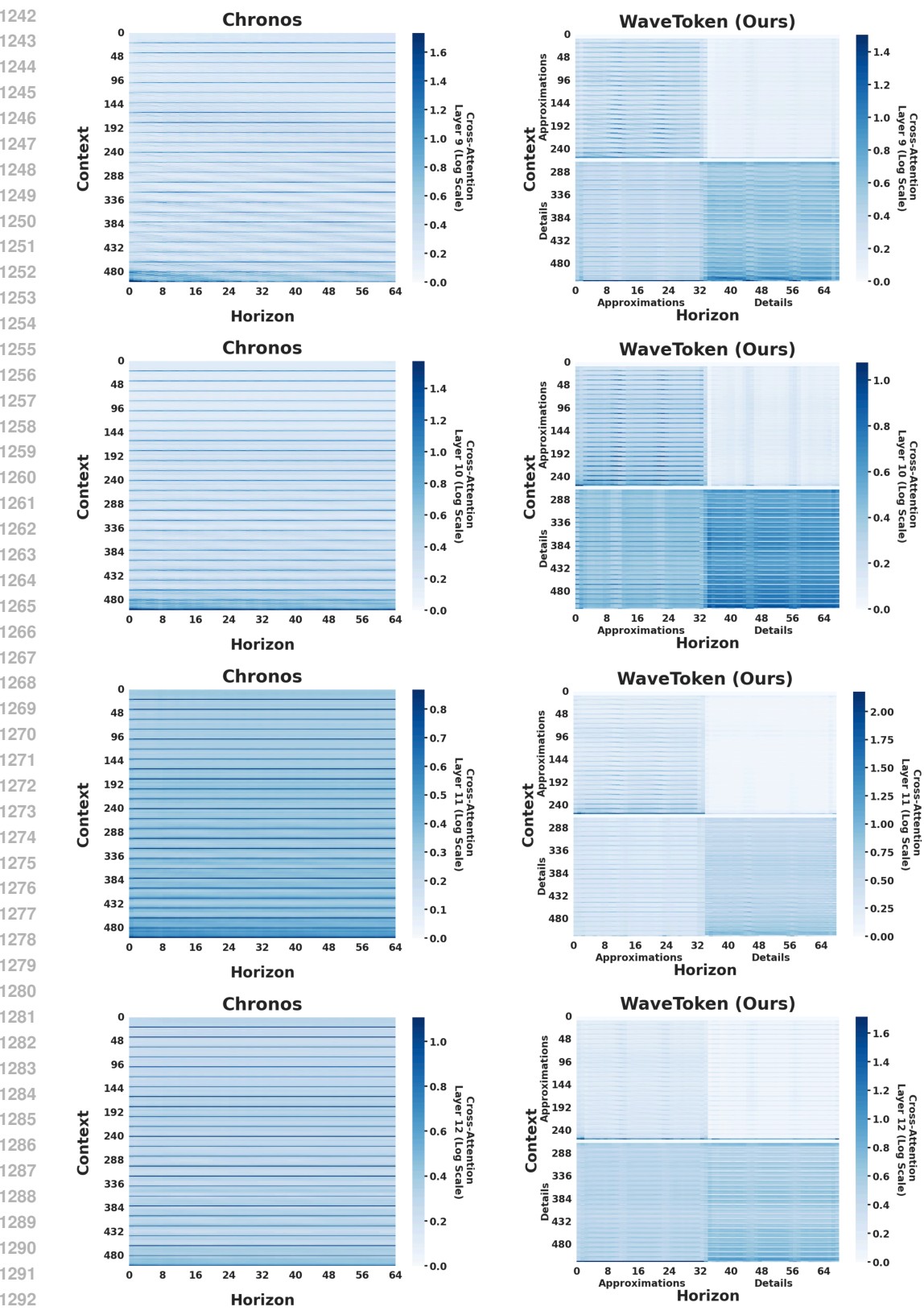

Figure 15: Cross-attention maps for the ninth to twelfth (from top) decoder layers of Chronos-Base (left) and WaveToken-Base (right) when forecasting the spiky data of the second row in Figure 1.

Table 9: Details of all datasets used for experiments, as collected by Ansari et al. (2024), partitioned according to how they are used for training and evaluation of WaveToken models.

| Dataset | Domain | Freq. | Num. Series | Series Length | | | Prediction |
|---|---|---|---|---|---|---|---|
| | | | | min | avg | max | Length ($H$) |
| **Pretraining-only** | | | | | | | |
| Solar (5 Min.) | energy | 5min | 5166 | 105120 | 105120 | 105120 | - |
| Solar (Hourly) | energy | 1h | 5166 | 8760 | 8760 | 8760 | - |
| Spanish Energy and Weather | energy | 1h | 66 | 35064 | 35064 | 35064 | - |
| Taxi (Hourly) | transport | 1h | 2428 | 734 | 739 | 744 | - |
| USHCN | nature | 1D | 6090 | 5906 | 38653 | 59283 | - |
| Weatherbench (Daily) | nature | 1D | 225280 | 14609 | 14609 | 14610 | - |
| Weatherbench (Hourly) | nature | 1h | 225280 | 350633 | 350639 | 350640 | - |
| Weatherbench (Weekly) | nature | 1W | 225280 | 2087 | 2087 | 2087 | - |
| Wiki Daily (100k) | web | 1D | 100000 | 2741 | 2741 | 2741 | - |
| Wind Farms (Daily) | energy | 1D | 337 | 71 | 354 | 366 | - |
| Wind Farms (Hourly) | energy | 1h | 337 | 1715 | 8514 | 8784 | - |
| **In-domain evaluation** | | | | | | | |
| Electricity (15 Min.) | energy | 15min | 370 | 16032 | 113341 | 140256 | 24 |
| Electricity (Hourly) | energy | 1h | 321 | 26304 | 26304 | 26304 | 24 |
| Electricity (Weekly) | energy | 1W | 321 | 156 | 156 | 156 | 8 |
| KDD Cup 2018 | nature | 1h | 270 | 9504 | 10897 | 10920 | 48 |
| London Smart Meters | energy | 30min | 5560 | 288 | 29951 | 39648 | 48 |
| M4 (Daily) | various | 1D | 4227 | 107 | 2371 | 9933 | 14 |
| M4 (Hourly) | various | 1h | 414 | 748 | 901 | 1008 | 48 |
| M4 (Monthly) | various | 1M | 48000 | 60 | 234 | 2812 | 18 |
| M4 (Weekly) | various | 1W | 359 | 93 | 1035 | 2610 | 13 |
| Pedestrian Counts | transport | 1h | 66 | 576 | 47459 | 96424 | 48 |
| Rideshare | transport | 1h | 2340 | 541 | 541 | 541 | 24 |
| Taxi (30 Min.) | transport | 30min | 2428 | 1469 | 1478 | 1488 | 48 |
| Temperature-Rain | nature | 1D | 32072 | 725 | 725 | 725 | 30 |
| Uber TLC (Daily) | transport | 1D | 262 | 181 | 181 | 181 | 7 |
| Uber TLC (Hourly) | transport | 1h | 262 | 4344 | 4344 | 4344 | 24 |
| **Zero-shot evaluation** | | | | | | | |
| Australian Electricity | energy | 30min | 5 | 230736 | 231052 | 232272 | 48 |
| CIF 2016 | banking | 1M | 72 | 28 | 98 | 120 | 12 |
| Car Parts | retail | 1M | 2674 | 51 | 51 | 51 | 12 |
| Covid Deaths | healthcare | 1D | 266 | 212 | 212 | 212 | 30 |
| Dominick | retail | 1D | 100014 | 201 | 296 | 399 | 8 |
| ERCOT Load | energy | 1h | 8 | 154854 | 154854 | 154854 | 24 |
| ETT (15 Min.) | energy | 15min | 14 | 69680 | 69680 | 69680 | 24 |
| ETT (Hourly) | energy | 1h | 14 | 17420 | 17420 | 17420 | 24 |
| Exchange Rate | finance | 1B | 8 | 7588 | 7588 | 7588 | 30 |
| FRED-MD | economic | 1M | 107 | 728 | 728 | 728 | 12 |
| Hospital | healthcare | 1M | 767 | 84 | 84 | 84 | 12 |
| M1 (Monthly) | various | 1M | 617 | 48 | 90 | 150 | 18 |
| M1 (Quarterly) | various | 3M | 203 | 18 | 48 | 114 | 8 |
| M1 (Yearly) | various | 1Y | 181 | 15 | 24 | 58 | 6 |
| M3 (Monthly) | various | 1M | 1428 | 66 | 117 | 144 | 18 |
| M3 (Quarterly) | various | 3M | 756 | 24 | 48 | 72 | 8 |
| M3 (Yearly) | various | 1Y | 645 | 20 | 28 | 47 | 6 |
| M4 (Quarterly) | various | 3M | 24000 | 24 | 100 | 874 | 8 |
| M4 (Yearly) | various | 1Y | 23000 | 19 | 37 | 841 | 6 |
| M5 | retail | 1D | 30490 | 124 | 1562 | 1969 | 28 |
| NN5 (Daily) | finance | 1D | 111 | 791 | 791 | 791 | 56 |
| NN5 (Weekly) | finance | 1W | 111 | 113 | 113 | 113 | 8 |
| Tourism (Monthly) | various | 1M | 366 | 91 | 298 | 333 | 24 |
| Tourism (Quarterly) | various | 1Q | 427 | 30 | 99 | 130 | 8 |
| Tourism (Yearly) | various | 1Y | 518 | 11 | 24 | 47 | 4 |
| Traffic | transport | 1h | 862 | 17544 | 17544 | 17544 | 24 |
| Weather | nature | 1D | 3010 | 1332 | 14296 | 65981 | 30 |

