# OpenReview forum: "Enhancing Foundation Models for Time Series Forecasting via Wavelet-based Tokenization"
_ICLR.cc/2025/Conference — Submitted to ICLR 2025_

### Official Review · Reviewer_qSp1 · 2024-10-26

**Soundness:** 3
**Presentation:** 3
**Contribution:** 2
**Rating:** 5
**Confidence:** 4

**Summary:**

The paper introduces WaveToken, a tokeniser that equips a time series model with wavelet-based features for uni-variate forecasting applications. The authors adapt an existing Transformer-based forecasting model, namely Chronos, by extracting features from a discrete wavelet transform during tokenisation. Evaluations in both in-distribution and zero-shot settings indicate that the use of wavelet-based features during tokenisation might be benfecial across uni-variate forecasting applications.

**Strengths:**

1. The paper is well structured.
2. The paper is clearly written.
3. The authors conduct extensive forecasting experiments, including 15 in-distribution and 27 zero-shot applications, to evaluate their method.
4. The authors perform a simple qualitative analysis using attention maps and conduct ablation studies to provide insights into their approach.

**Weaknesses:**

1. Time series models utilising wavelet-based features have been well studied for time series analysis [1,2,3,4,5,6]. As the work under review combines wavelet-based feature extraction with an existing time series model, namely Chronos, for a very specific task, i.e. uni-variate forecasting, it offers a limited technical contribution to the field of time series analysis.

2. In their qualitative analysis and their conclusion, the authors claim "excellent forecasting accuracy" as well as "superior generalization capabilities". However, the results in Figure 3 and 4 reveal that using wavelet-based features does not improve downstream performance with respect to weighted quantile loss (WQL) and mean absolute scaled error (MASE), in both in-distribution and zero-shot settings. In the former setting, wavelet-based tokenisation even hurts performance compaired to the adapted Chronos model. Only in the newly introduced visual relative squared error (VRSE) [7], the results indicate incremental improvements compared to the baselines.

3. The authors only analyse wavelet-based features for autoregressive models, i.e. Chronos. It would be worth exploring WaveToken in combination with other forecasting models, e.g. non-autoregressive models, or even general time series models. This would provide detailed insight whether wavelet-based features may be beneficial for other architectures or for other tasks such as classification, regression, or imputation.

4. The authors do not report their results across multiple seeds to ensure robustness.

5. The authors do not elaborate on the limitations of their work.

6. The authors neither provide configurations for hyperparamter tuning (including the final hyperparameter setting) nor share their code to support reproducibility.

7. The terminology of the paper lacks clarity. For instance, the authors use the term "language model" throughout their work, which might be confusing since the model is trained from scratch on a large collection of time series. The term time series model would be more appropriate. Additionally, the authors alternate between forecasting "accuracy" and "performance", with the latter being the more suitable term. It should be used consistently throughout the paper to avoid confusion. While technical terms such as "time-localized frequencies" (ll. 14-15) may be familiar in the signal processing community, a brief explanation would help make these concepts accessible to a broader time series audience. In line 415, the authors state "Both Chronos, TimesFM, and Moirai" where I believe they intended to say "~~Both~~ Chronos, TimesFM, and Moirai".

[1] Torrence et al. "A practical guide to wavelet analysis." Bulletin of the American Meteorological Society (1998).

[2] Percival, D. B. "Wavelet Methods for Time Series Analysis." Cambridge University Press (2000).

[3] Cazelles et al. "Wavelet analysis of ecological time series." Oecologia (2008).

[4] Pandey et al. "Intelligent hybrid wavelet models for short-term load forecasting." IEEE Transactions on Power Systems (2010).

[5] Wang et al. "Multilevel wavelet decomposition network for interpretable time series analysis." ACM SIGKDD International Conference on Knowledge Discovery & Data Mining (2018).

[6] Sasal et al. "W-transformers: a wavelet-based transformer framework for univariate time series forecasting." IEEE international conference on machine learning and applications (2022).

[7] Posam et al. "DiffFind: Discovering Differential Equations from Time Series." Pacific-Asia Conference on Knowledge Discovery and Data Mining. Springer Nature Singapore (2024).

**Questions:**

1. Have the authors analysed the learned vocabulary? Are there any patterns present in the vocabulary that can be leveraged to further improve tokenisation and ideally downstream performance?

2. The authors state that existing time series models which preserve the natural temporal dependency "tend to focus more either on the recent history or uniformly over all time steps" (ll. 78-79), however, without providing any ground for this claim. Existing time series models [8,9,10,11,12] employ attention-based Transformers, that are known for their attention-weighted (i.e., non-uniform) focus on time steps. Could the authors therefore elaborate further on their statement?

3. How do the authors determine the "average rank" (l. 395) reported e.g. in Figure 10?

4. The authors state that the cross-entropy loss "lends itself well to forecasting applications, where it is often important to capture the correct shape or pattern of the time series without imposing structural limitations such as those inherent in traditional loss functions like MSE and MAE" (ll. 295-298). Could the authors elaborate further on this statement?

5. Why would the authors "use the last $H$ observations of each time series as a held-out test set" (l. 357) even for zero-shot benchmarking? Are they training or evaluating on the remaining time steps? Why not evaluate the entire time series in the established rolling window fashion [13]?

[8] Ansari et al. "Chronos: Learning the language of time series." arXiv (2024).

[9] Woo et al. "Unified training of universal time series forecasting transformers." ICML (2024).

[10] Goswami et al. "Moment: A family of open time-series foundation models." ICML (2024).

[11] Yang et al. "Biot: Biosignal transformer for cross-data learning in the wild." NeurIPS (2024).

[12] Jiang et al. "Large brain model for learning generic representations with tremendous EEG data in BCI." ICLR (2024).

[13] Zhou et al. "Informer: Beyond efficient transformer for long sequence time-series forecasting." AAAI (2021).

---

> ### Author Response · Authors · 2024-11-21
> **OFFICIAL RESPONSE TO REVIEWER qSp1**
>
> Thank you for your useful comments and suggestions that helped improve our paper. We have summarized our main revisions and answers in response to all reviewers in a separate comment above. Below we provide point-by-point answers to each section in your review.
>
> ---
> Regarding
> > In their qualitative analysis and their conclusion, the authors claim "excellent forecasting accuracy" as well as "superior generalization capabilities". However, the results in Figure 3 and 4 reveal that using wavelet-based features does not improve downstream performance with respect to weighted quantile loss (WQL) and mean absolute scaled error (MASE), in both in-distribution and zero-shot settings. In the former setting, wavelet-based tokenisation even hurts performance compaired to the adapted Chronos model. Only in the newly introduced visual relative squared error (VRSE) [7], the results indicate incremental improvements compared to the baselines.
>
> The *core motivation of our work* was to develop a general purpose tokenizer that seamlessly captures global and local patterns in the time series. Our **qualitative results** (Figure 1 and Section 4.3) demonstrate the impressive performance of WaveToken on complex edge cases where existing foundation models fail, almost completely. These results highlight the remarkable ability of a wavelet-based tokenizer to capture nonstationary signals – which are pervasive in practical applications. Nevertheless, as outlined in the paper, **WaveToken also improves quantitatively** on Chronos 75% of the times and 83% of the times on the in-domain (15 datasets) and zero-shot (27 datasets) benchmarks, respectively, while employing a vocabulary one-quarter the size of Chronos. In addition, *WaveToken achieves the best average rank across all metrics* for both Benchmarks I and II (Figures 10 and 11).
>
> We believe that in the natural progression of the field, different works make different types of contributions, and not every work brings (or needs to bring) a paradigm-shifting accuracy improvement. In the case of WaveToken, our findings position it as a promising avenue for developing general-purpose, information-efficient forecasting models, and it should therefore be evaluated independently in addition to its accuracy.
>
> ---
> Regarding
> > The authors do not report their results across multiple seeds to ensure robustness.
>
> All results for WaveToken and Chronos are reported by averaging metrics across 3 different training runs (as mentioned in Appendix D), in order to draw reliable and reproducible conclusions. In hindsight, we realize we should have mentioned this in the main text as well. We have updated our manuscript accordingly.
>
> ---
> Regarding
> > The authors do not elaborate on the limitations of their work.
>
> Thanks for pointing this out, we have added the paragraph below at the end of Section 5.
>
> Specifically, WaveToken exploits an autoregressive model on wavelet coefficients. As such, it suffers from slower decoding and inference time relative to recently proposed alternatives, such as patch-based models. Applying WaveToken to patch-based architectures represents an interesting area for future research.
>
> ---
> Regarding
> > The authors neither provide configurations for hyperparamter tuning (including the final hyperparameter setting) nor share their code to support reproducibility
>
> Section 4.1 provides all the details on our model, including parameter counts, architecture used, context and prediction length, training and evaluation strategies. In addition, Appendix F reports all the hyper-parameters used for each model, including all the baselines.
>
> As for code, we plan to release a user-friendly research package complete with details on how to train and evaluate our tokenizer and models. Unfortunately, we could not share our code for review at this stage due to pending legal approvals.
>
> ---
> Regarding
> > How do the authors determine the "average rank" (l. 395) reported e.g. in Figure 10?
>
> The average rank is computed by calculating the rank of each model on each dataset according to its forecasting performance relative to the other baselines, for a specific metric. The ranks are then averaged across all the datasets within a benchmark, i.e. separately for in-domain and zero-shot datasets.

---

> > ### Author Response · Authors · 2024-11-21
> > **OFFICIAL RESPONSE TO REVIEWER qSp1 (cont.)**
> >
> > Regarding
> > > The authors state that the cross-entropy loss "lends itself well to forecasting applications, where it is often important to capture the correct shape or pattern of the time series without imposing structural limitations such as those inherent in traditional loss functions like MSE and MAE" (ll. 295-298). Could the authors elaborate further on this statement?
> >
> > While unconventional, using the cross-entropy loss for continuous data is not a new idea and is known in the literature as **regression via classification** [1, 2]. Previous works have shown the benefits of this approach across tasks, such as audio modeling (Wavenet [3]), Deep RL [4], and recently in the context of forecasting (Chronos [5]). Although the cross-entropy loss removes topological information, the model implicitly learns the topology from the data through large scale training. Using a categorical distribution allows the model to be flexible to accommodate multimodalities, strong asymmetries, sudden shifts, and more, as the distribution is not constrained to a specific shape.
> >
> > Additionally, the use of cross-entropy loss allows to sidestep common issues with standard forecasting objectives (like MSE or quantile loss) such as their dependence on the scale of the inputs and on the presence of outliers, which can significantly degrade learning. Conversely, other approaches – e.g., DeepAR or more recent foundation models like Moirai – impose restrictive assumptions on the output forecasts via parametric distributions (usually Gaussian or Student’s-t), which are often not empirically valid. In our comparisons with these models, we already compare WaveToken against these alternative objective functions.
> >
> > [1] Torgo, L., & Gama, J. (1997). Regression using classification algorithms. Intelligent Data Analysis, 1(4), 275-292
> >
> > [2] Stewart, L., Bach, F., Berthet, Q., & Vert, J. P. (2023, April). Regression as classification: Influence of task formulation on neural network features. In International Conference on Artificial Intelligence and Statistics (pp. 11563-11582). PMLR
> >
> > [3] Van den Oord A., Dieleman S., Zen H., Simonyan K., Vinyals O., Graves A., Kalchbrenner N., Senior A., Kavukcuoglu K. (2016), WaveNet: A Generative Model for Raw Audio, arXiv preprint arXiv:1609.03499
> >
> > [4] Farebrother, J., Orbay, J., Vuong, Q., Taïga, A. A., Chebotar, Y., Xiao, T., ... & Agarwal, R. (2024). Stop regressing: Training value functions via classification for scalable deep rl. arXiv preprint arXiv:2403.03950
> >
> > [5] Ansari, A. F., Stella, L., Turkmen, C., Zhang, X., Mercado, P., Shen, H., ... & Wang, Y. (2024). Chronos: Learning the language of time series. arXiv preprint arXiv:2403.07815
> >
> > ---
> > Regarding
> > > Why would the authors "use the last H observations of each time series as a held-out test set" (l. 357) even for zero-shot benchmarking? Are they training or evaluating on the remaining time steps? Why not evaluate the entire time series in the established rolling window fashion [13]?
> >
> > We follow the evaluation protocol reported in the Chronos paper to provide a fair evaluation of our wavelet-based tokenizer. Similar protocols have been used in other works [6, 7, 8]. While rolling evaluation may help provide accurate performance estimates when fewer datasets are involved (e.g., in [13] which uses 4 datasets), our larger scale evaluation on 42 datasets is comprehensive enough to provide an accurate aggregate performance of the models even with a single evaluation window per task. Furthermore, the evaluation protocols for several datasets are a standard in the forecasting community as they have been derived from the respective competitions. These include M1, M3, M4, M5, KKD Cup 2018, NN5 and CIF 2016 competition datasets.
> >
> > [6] Gruver N., Finzi M., Qiu S., Wilson A.G. (2023), Large language models are zero-shot time series forecasters
> >
> > [7] Das A., Kong W., Sen R., Zhou Y. (2023), A decoder-only foundation model for time-series forecasting
> >
> > [8] Godahewa R., Bergmeir C., Webb G.I., Hyndman R.J., Montero-Manso P. (2021), Monash Time Series Forecasting Archive

---

> ### Author Response · Authors · 2024-11-25
> **A Gentle Reminder**
>
> Dear Reviewer qSp1,
>
> Thank you again for the useful comments and questions that helped improve our paper. As the end of the discussion period is approaching, we would like to gently remind you to take into account our responses and the updated version of the manuscript into your evaluation.
>
> We hope that we have satisfactorily addressed your questions and concerns. If so, we request you to consider raising your score to reflect that. If you have further questions, we will be happy to respond to them.
>
> Thank you.
>
> Best Regards,
> The Authors

---

> ### Author Response · Authors · 2024-11-27
> **Follow-up Reminder**
>
> Dear Reviewer qSp1,
>
> Thank you again for your review. We kindly remind you to take into account our responses and the updated version of the manuscript into your evaluation.
>
> We hope that we have satisfactorily addressed your questions and concerns. If so, we request you to consider raising your score to reflect that. If you have further questions, we will be happy to respond to them.
>
> Thank you.
>
> Best Regards,\
> The Authors

---

> ### Author Response · Authors · 2024-11-30
> **Reminder before End of Discussion Period**
>
> Dear Reviewer qSp1,
>
> Thank you again for your helpful review. As you know, the discussion period will end in two days on Monday December 2nd. We would like to gently remind you to take into account our responses to your questions and concerns.
>
> We hope our answers above have satisfactorily addressed your comments. If so, we request you to consider raising your score to reflect that. If you have further questions, we would be happy to respond to them.
>
> Thank you.

---

> ### Comment · Area_Chair_XhKp · 2024-12-02
> **Please respond to the authors of submission 8429**
>
> Dear Reviewer qSp1,
>
> The authors of submission 8429 have provided an extensive response to your review.
>
> As the discussion period is almost over, please go over their response and determine which of your comments are addressed.
>
> Please explain your decision to update (or not update) your score.
>
> All the best,
>
> The AC

---

> ### Comment · Reviewer_qSp1 · 2024-12-03
> **Reviewer Response**
>
> Thank you very much for the response and the revised manuscript. I have carefully read both the rebuttal and the updated version of the study.
>
> The authors have conducted an interesting investigation on whether wavelet-based features can enhance time series foundation models, such as Chronos [1].
>
> While I find the contribution of this work to be limited, given that wavelet-based approaches have been extensively studied in the literature, including the field of time series analysis [2][3][4][5][6][7][8], this would not be a concern if the method demonstrated clear benefits for downstream performance. However, the experimental results indicate that the proposed approach does not offer clear advantages for downstream applications, especially when compared to the model it builds upon, i.e. Chonos [1].
>
> Additionally, the rebuttal does not address my points related to other non-regressive forecasting models or general time series models, which could potentially benefit from wavelet-based features. The study also lacks an analysis of the learned vocabulary to explore whether patterns can be leveraged to improve the downstream performance.
>
> Given these considerations, I cannot recommend accepting the study at this time, and thus leave my scores unchanged. However, I encourage the authors to refine and expand upon this work, as the conceptual idea of wavelet-based tokenisation is interesting.
>
> ---
> [1] Ansari et al. "Chronos: Learning the language of time series." arXiv (2024).
>
> [2] Torrence et al. "A practical guide to wavelet analysis." Bulletin of the American Meteorological Society (1998).
>
> [3] Percival, D. B. "Wavelet Methods for Time Series Analysis." Cambridge University Press (2000).
>
> [4] Cazelles et al. "Wavelet analysis of ecological time series." Oecologia (2008).
>
> [5] Pandey et al. "Intelligent hybrid wavelet models for short-term load forecasting." IEEE Transactions on Power Systems (2010).
>
> [6] Wang et al. "Multilevel wavelet decomposition network for interpretable time series analysis." ACM SIGKDD International Conference on Knowledge Discovery & Data Mining (2018).
>
> [7] Sasal et al. "W-transformers: a wavelet-based transformer framework for univariate time series forecasting." IEEE international conference on machine learning and applications (2022).
>
> [8] Posam et al. "DiffFind: Discovering Differential Equations from Time Series." Pacific-Asia Conference on Knowledge Discovery and Data Mining. Springer Nature Singapore (2024).

---

> ### Author Response · Authors · 2024-12-03
>
> Thank you for your response and follow-up comments.
>
> We agree with the reviewer that wavelet-based methods have been used previously in the context of time series forecasting. However, WaveToken uses it in the completely new context of tokenization for time series foundation models. These foundation models, such as Chronos, have received considerable attention from the time series community of late and our work addresses a critical component (tokenization) of these models.
>
> We respectfully disagree with reviewer's assertion that "the proposed approach does not offer clear advantages for downstream applications". On the contrary, WaveToken improves quantitatively over Chronos 75% of the times and 83% of the times on the in-domain (15 datasets) and zero-shot (27 datasets) benchmarks, respectively. In the long-horizon experiments that we have added to the revision (Fig. 12) too, WaveToken shows its edge over Chronos. Furthermore, in the qualitative analysis (Figure 1 and Section 4.3), the difference between WaveToken and Chronos is stark, where Chronos completely fails on spiky and non-stationary data.
>
> > Additionally, the rebuttal does not address my points related to other non-regressive forecasting models or general time series models, which could potentially benefit from wavelet-based features.
>
> Indeed, wavelet-based features could potentially help other time series models. However, our work is NOT about wavelet features in general, but about _wavelet-based tokenization_ which is applicable to autoregressive methods that employ tokenization (such as Chronos) and not directly relevant to non-autoregressive models.
>
> > The study also lacks an analysis of the learned vocabulary to explore whether patterns can be leveraged to improve the downstream performance.
>
> It is unclear what kind of analysis the reviewer is referring to here. In the context of WaveToken, the vocabulary are the bins of the wavelet coefficients obtained upon quantization. As such, the vocabulary is neither learned nor it is a vocabulary of patterns.
>
> We hope that the reviewer reconsiders their score in the light of our response above.

---

### Official Review · Reviewer_KBzW · 2024-10-27

**Soundness:** 3
**Presentation:** 2
**Contribution:** 2
**Rating:** 6
**Confidence:** 3

**Summary:**

This paper aims to develop foundation models for time series forecasting. The authors have delved into the tokenization approach for real-valued time series inputs and propose WaveToken, a wavelet-based tokenizer that involves scaling, decomposing, and quantizing the wavelet coefficients on time series. An autoregressive language model is adapted to generate future tokens of wavelet coefficients. Experimental results demonstrate that the model performs well across diverse datasets in both in-distribution and zero-shot settings, using a smaller vocabulary compared with the previous tokenization method.

**Strengths:**

1. The paper proposes a novel tokenization from the perspective of the frequency domain.
2. The model is pre-trained on a large corpus of diverse datasets, with evaluations showing its effectiveness in capturing local and global patterns.

**Weaknesses:**

1. There are some overclaims: WaveToken claimed to be a foundation model, however, being limited to a fixed context length is a pronounced defect.
2. Limited refinement over previous works: From the experimental results, the performance of the proposed model is not significant. The overall promotions (Table 3 and Table 4) are close to that of Chronos. Besides, the model seems to have a disadvantage in efficiency compared with the straightforward quantizing tokenization of Chronos.
3. Learning autoregressive models on the wavelet coefficients, which do not preserve the temporal transitions, does not make sense to me. The author also noticed the counter-intuition. However, the insignificance of experimental results and lack of theoretical proof amplifies this irrationality. Can the author give more explanations and proofs in this respect? I am not denying the wavelet tokens, which have been extensively explored in previous works before. It might be more reasonable to learn the diffusion process on the decomposition-based coefficients. Whether the author has tried in this regard?
4. About the experiments to support wavelet tokenization: As previously stated, the main evidence in the paper supporting wavelet tokenization comes from the performance of in-domain and zero-shot generalization. However, recent work has shown that there is a risk of overfitting these smaller time series datasets with the LLM [1]. It is recommended that the author provide more comparisons on diverse benchmarks.
5. Being able to use a smaller vocabulary does not seem like a particularly appealing advantage at present. Could the authors provide more results to verify whether the wavelet-based vocabulary can be extremely small?

[1] Are Language Models Actually Useful for Time Series Forecasting?

**Questions:**

1. How does WaveToken perform in long-term forecasting? Recent work has shown that Chronos can be less effective at predicting long horizons than models that adopt patch tokens (such as TimesFM). It may be due to the multi-step error accumulation caused by its point-wise tokenization. I wonder if WaveToken can effectively solve this problem.
2. WaveToken and Chronos both adopt the LLM-style discrete vocabulary, which can naturally support probabilistic forecasting. Meanwhile, deep forecasters using continuous embeddings (linear layers or MLPs) can also support it by introducing a probabilistic head (such as TimesFM and Moirai). What other advantages do you think discrete vocabularies have over continuous embeddings in time series forecasters?

---

> ### Author Response · Authors · 2024-11-21
> **OFFICIAL RESPONSE TO REVIEWER KBzW**
>
> Thank you for your comments and questions that helped improve our paper. We have summarized our main revisions and answers in response to all reviewers in a separate comment above. Below we provide point-by-point answers to each section in your review.
>
> ---
> Regarding
> > There are some overclaims: WaveToken claimed to be a foundation model, however, being limited to a fixed context length is a pronounced defect.
>
> WaveToken is only **trained** with a fixed context and prediction length, but during inference one is free to use arbitrary context and prediction lengths. This setup is not special to WaveToken and is also used by other foundation models for forecasting such as Chronos and TimesFM. We opted for training context and prediction lengths of 512 and 64, respectively, as they are sufficient for many practical use cases, as shown by our extensive evaluation on 42 datasets that cover different history lengths, prediction lengths, seasonalities, and domains.
>
> ---
> Regarding
> > Limited refinement over previous works: From the experimental results, the performance of the proposed model is not significant. The overall promotions (Table 3 and Table 4) are close to that of Chronos. Besides, the model seems to have a disadvantage in efficiency compared with the straightforward quantizing tokenization of Chronos.
>
> Firstly, as noted in our response to a similar comment from reviewer zVkm, the *core motivation of our work* was to develop a general purpose tokenizer that seamlessly captures global and local patterns in the time series. Our **qualitative results** (Figure 1 and Section 4.3) demonstrate the impressive performance of WaveToken on complex edge cases where existing foundation models fail, almost completely. These results highlight the remarkable ability of a wavelet-based tokenizer to capture nonstationary signals – which are pervasive in practical applications. Nevertheless, as outlined in the paper, **WaveToken also improves quantitatively** on Chronos 75% of the times and 83% of the times on the in-domain (15 datasets) and zero-shot (27 datasets) benchmarks, respectively, while employing a vocabulary one-quarter the size of Chronos. In addition, WaveToken achieves the *best average rank across all metrics for both Benchmarks I and II* (Figures 10 and 11). Note that our comparison with Chronos, which uses a simple uniform quantization scheme directly on the time series values, readily serves as an ablation against a simpler tokenization scheme (the simplest possible scheme in this context).
>
> We believe that in the natural progression of the field, different works make different types of contributions, and not every work brings (or needs to bring) a paradigm-shifting accuracy improvement. In the case of WaveToken, our findings position it as a promising avenue for developing general-purpose, information-efficient forecasting models, and it should therefore be evaluated independently in addition to its accuracy.
>
> Secondly, *WakeToken does not have a disadvantage in terms of* **efficiency** as the wavelet tokenization is a convolution-and-downsampling operation which runs in linear time and the inference speed differences are not perceivable even when the wavelet transform is done in the CPU, as in our current implementation. Below we report running times (averaged over 10 repetitions) for both Chronos (Base) and WaveToken (Base) when forecasting a batch of 32 time series, with context length equal to 512 and prediction length equal to 64:
> - Chronos: 6.56s +- 1.02ms
> - WaveToken: 6.86s +- 1.14ms
>
> The difference in inference times is negligible, and could be further reduced by applying the wavelet decomposition in parallel on the GPU.

---

> ### Author Response · Authors · 2024-11-21
> **OFFICIAL RESPONSE TO REVIEWER KBzW (cont.)**
>
> Regarding
> > Learning autoregressive models on the wavelet coefficients, which do not preserve the temporal transitions, does not make sense to me. The author also noticed the counter-intuition. However, the insignificance of experimental results and lack of theoretical proof amplifies this irrationality. Can the author give more explanations and proofs in this respect? I am not denying the wavelet tokens, which have been extensively explored in previous works before. It might be more reasonable to learn the diffusion process on the decomposition-based coefficients. Whether the author has tried in this regard?
>
> To clarify, *temporal dependencies are preserved within each coefficient group*. In other words, with a one-level wavelet decomposition, WaveToken is simply providing additional structure (namely, high-frequency versus low-frequency coefficients, divided in two concatenated groups) that the attention-based mechanism can leverage. The autoregressive model is still consistent within each coefficient group, and the attention mechanism allows it to determine when to switch from an autoregressive regime appropriate for low frequencies to an autoregressive regime appropriate for high frequencies.
>
> This phenomenon is also highlighted in the *attention maps presented in Figure 5*: by decomposing the input sequence into approximation (low-frequency) and detail (high-frequency) coefficients, the model learns more easily to focus on the important portions to effectively forecast complex time series of practical relevance such as sparse spikes or non-stationary frequencies evolving over time, as can be seen in Figure 1.
>
> ---
> Regarding
> > About the experiments to support wavelet tokenization: As previously stated, the main evidence in the paper supporting wavelet tokenization comes from the performance of in-domain and zero-shot generalization. However, recent work has shown that there is a risk of overfitting these smaller time series datasets with the LLM [1]. It is recommended that the author provide more comparisons on diverse benchmarks.
>
> WaveToken is trained on a large corpus of real and synthetic time series from diverse sources, NOT small individual time series datasets as studied in [1]. Our remarkable zero-shot results on 27 datasets showing that WaveToken performs better or on par with models trained on these datasets (e.g., DeepAR, TFT, PatchTST), despite never having seen these datasets, demonstrates that the risk of overfitting is non-existent. Furthermore, the models and evaluation settings considered in [1] have been superseded by the large-scale zero-shot evaluations in other recent works such as Chronos and TimesFM.
>
> [1] Tan, M., Merrill, M. A., Gupta, V., Althoff, T., & Hartvigsen, T. (2024, June). Are language models actually useful for time series forecasting?. In The Thirty-eighth Annual Conference on Neural Information Processing Systems.
>
> ---
> Regarding
> > How does WaveToken perform in long-term forecasting? Recent work has shown that Chronos can be less effective at predicting long horizons than models that adopt patch tokens (such as TimesFM). It may be due to the multi-step error accumulation caused by its point-wise tokenization. I wonder if WaveToken can effectively solve this problem.
>
> Thank you for suggesting this additional setting. We conducted **new experiments** (see the new Figure 12 and Appendix E in the paper) on a **long-horizon benchmark** constructed by increasing the forecast length $H$ of each dataset in Benchmark II (Zero-Shot) by a factor of 2 and 3 (see Table 9 for original forecast lengths). The datasets in Benchmark II without sufficient history in all time series to allow for longer horizons have been skipped. **WaveToken outperforms other foundation models across all three metrics** in the $H \times 2$ setting. When $H \times 3$, TimesFM performs better with respect to WQL only. All results for Chronos and WaveToken are averaged across three different seeds.

---

> > ### Author Response · Authors · 2024-11-21
> > **OFFICIAL RESPONSE TO REVIEWER KBzW (cont.)**
> >
> > Regarding
> > > WaveToken and Chronos both adopt the LLM-style discrete vocabulary, which can naturally support probabilistic forecasting. Meanwhile, deep forecasters using continuous embeddings (linear layers or MLPs) can also support it by introducing a probabilistic head (such as TimesFM and Moirai). What other advantages do you think discrete vocabularies have over continuous embeddings in time series forecasters?
> >
> > A **discrete vocabulary**, in conjunction with the categorical cross-entropy loss, offers the following advantages:
> > - No assumptions are made about the shape of the distribution and the model is free to learn and generate arbitrary distributions. As such, this makes the model flexible to accommodate multimodalities, strong asymmetries, sudden shifts, and more. This is especially critical in the context of a pretrained model such as WaveToken which may be used with arbitrary downstream datasets.
> > - It allows the model to sidestep common issues with standard forecasting objectives (like MSE or Quantile loss) such as their dependence on the scale of the inputs and on the presence of outliers, which can significantly degrade learning.
> >
> > Nevertheless, a discrete vocabulary loses the topological information of the continuous domain. However, this is a limited cause of concern as the model learns this information from the data through large scale training as shown by WaveToken’s empirical performance.

---

> ### Author Response · Authors · 2024-11-25
> **A Gentle Reminder**
>
> Dear Reviewer KBzW,
>
> Thank you again for the useful comments and questions that helped improve our paper. As the end of the discussion period is approaching, we would like to gently remind you to take into account our responses and the updated version of the manuscript into your evaluation.
>
> We hope that we have satisfactorily addressed your questions and concerns. If so, we request you to consider raising your score to reflect that. If you have further questions, we will be happy to respond to them.
>
> Thank you.
>
> Best Regards,
> The Authors

---

> ### Comment · Reviewer_KBzW · 2024-11-25
>
> Thank you for the detailed response and additional empirical experiments, which addressed my concerns regarding the performance brought by wavelet tokenization and the difference from previous LLM-based time series models. The results of long-term forecasting are inspiring. The response regarding the discussion of discrete/continuous tokenization is convincing to me, which highlights that discrete vocabulary can be more adaptive to the distributions of time series. Thus, the proposed wavelet-based approach may shield light on this community.
>
> After reading all the comments and responses, I'd like to raise my score to the acceptance level.

---

### Official Review · Reviewer_uEP1 · 2024-10-30

**Soundness:** 3
**Presentation:** 4
**Contribution:** 3
**Rating:** 8
**Confidence:** 4

**Summary:**

This paper proposes a wavelet-based tokenization approach (WaveToken) to enhance time series forecasting using language models. It discusses one critical issue, i.e., tokenizing continuous, real-valued time series data, which hinders handling it using LLMs.

**Strengths:**

In overall, I agree with the importance of the problem. The paper is well-motivated and well-written. The mathematical formulations and methodological steps are sound.

**Weaknesses:**

Minor clarifications that could enhance the understanding and applicability of the approach.

**Questions:**

- The practical impact of the thresholding decisions on model generalization is uncertain. For example, aggressive thresholding could obscure subtle patterns, which could limit the model's robustness across diverse datasets. So, how do you ensure such an effect would not happen?
- Using cross-entropy loss for next-token prediction in a continuous-data domain is unconventional, as it aligns more with the categorical outputs. So, how it guides learning for continuous variables is still ambiguous for me. Is it possible to compare with other loss functions, e.g. MSE?
- Will the code be released?

---

> ### Author Response · Authors · 2024-11-21
> **OFFICIAL RESPONSE TO REVIEWER uEP1**
>
> Thank you for your useful comments and suggestions that helped improve our paper. We have summarized our main revisions and answers in response to all reviewers in a separate comment above. Below we provide point-by-point answers to each section in your review.
>
> ---
> Regarding
> > The practical impact of the thresholding decisions on model generalization is uncertain. For example, aggressive thresholding could obscure subtle patterns, which could limit the model's robustness across diverse datasets. So, how do you ensure such an effect would not happen?
>
> We did not explicitly threshold the coefficients as thresholding did not significantly improve the forecasting performance (see Section 4.4). As noted in lines 509-513, this can be attributed to the quantization step of the tokenizer: all wavelet coefficients close to 0 are mapped to the bin centered at 0. This value is then used at inference time to forecast tokens mapped to this bin, thereby *implicitly* thresholding small coefficients. Nevertheless, we agree that aggressive thresholding, or quantization in our context, may obscure subtle patterns. However, when coupled with instance normalization (i.e., standardization), this is a limited cause of concern on most real-world datasets, as demonstrated by the excellent quantitative and qualitative performance of WaveToken on 42 benchmark datasets.
>
> ---
> Regarding
> > Using cross-entropy loss for next-token prediction in a continuous-data domain is unconventional, as it aligns more with the categorical outputs. So, how it guides learning for continuous variables is still ambiguous for me. Is it possible to compare with other loss functions, e.g. MSE?
>
> While unconventional, using the cross-entropy loss for continuous data is not a new idea and is known in the literature as **regression via classification** [1, 2]. Previous works have shown the benefits of this approach across tasks, such as audio modeling (Wavenet [3]), Deep RL [4], and recently in the context of forecasting (Chronos [5]). Although the cross-entropy loss removes topological information, the model implicitly learns the topology from the data through large scale training. Using a categorical distribution allows the model to be flexible to accommodate multimodalities, strong asymmetries, sudden shifts, and more, as the distribution is not constrained to a specific shape.
>
> Additionally, the use of cross-entropy loss allows to sidestep common issues with standard forecasting objectives (like MSE or quantile loss) such as their dependence on the scale of the inputs and on the presence of outliers, which can significantly degrade learning. Conversely, other approaches – e.g., DeepAR or more recent foundation models like Moirai – impose restrictive assumptions on the output forecasts via parametric distributions (usually Gaussian or Student’s-t), which are often not empirically valid. In our comparisons with these models, we already compare WaveToken against these alternative objective functions.
>
> [1] Torgo, L., & Gama, J. (1997). Regression using classification algorithms. Intelligent Data Analysis, 1(4), 275-292
>
> [2] Stewart, L., Bach, F., Berthet, Q., & Vert, J. P. (2023, April). Regression as classification: Influence of task formulation on neural network features. In International Conference on Artificial Intelligence and Statistics (pp. 11563-11582). PMLR
>
> [3] Van den Oord A., Dieleman S., Zen H., Simonyan K., Vinyals O., Graves A., Kalchbrenner N., Senior A., Kavukcuoglu K. (2016), WaveNet: A Generative Model for Raw Audio, arXiv preprint arXiv:1609.03499
>
> [4] Farebrother, J., Orbay, J., Vuong, Q., Taïga, A. A., Chebotar, Y., Xiao, T., ... & Agarwal, R. (2024). Stop regressing: Training value functions via classification for scalable deep RL. arXiv preprint arXiv:2403.03950
>
> [5] Ansari, A. F., Stella, L., Turkmen, C., Zhang, X., Mercado, P., Shen, H., ... & Wang, Y. (2024). Chronos: Learning the language of time series. arXiv preprint arXiv:2403.07815
>
> ---
> Regarding
> > Will the code be released?
>
> Yes, we plan to release a user-friendly research package complete with details on how to train and evaluate our tokenizer and models. Unfortunately, we could not share our code for review at this stage due to pending legal approvals.

---

> ### Author Response · Authors · 2024-11-25
> **A Gentle Reminder**
>
> Dear Reviewer uEP1,
>
> Thank you again for the useful comments and questions that helped improve our paper. As the end of the discussion period is approaching, we would like to gently remind you to take into account our responses and the updated version of the manuscript into your evaluation.
>
> We hope that we have satisfactorily addressed your questions and concerns. If you have further questions, we will be happy to respond to them.
>
> Thank you.
>
> Best Regards,
> The Authors

---

### Official Review · Reviewer_zVkm · 2024-11-01

**Soundness:** 2
**Presentation:** 1
**Contribution:** 2
**Rating:** 3
**Confidence:** 3

**Summary:**

The paper introduces a novel tokenizer, WaveToken, to time series analysis via wavelet transforms. The method first normalizes input time series using z-score scaling, then applies a maximally decimated Discrete Wavelet Transform using the Biorthogonal-2.2 wavelet family. This decomposition separates signals into approximation coefficients (low-frequency) and detail coefficients (high-frequency) while preserving input length. The coefficients are quantized into discrete tokens using Freedman-Diaconis binning, creating a compact 1024-token vocabulary that captures both global and local patterns. The tokenized coefficients are concatenated and fed into a T5 encoder-decoder architecture trained via next-token prediction. During inference, the model autoregressively generates tokens representing future wavelet coefficients, which are then inverse-transformed to produce forecasts. The model shows strong zero-shot generalization across domains.

**Strengths:**

1. Introduces wavelet-based tokenizer which bridges classical signal processing with modern deep learning.
2. Gets comparable or better performance with a smaller vocabulary (1024 vs 4096 tokens), demonstrating better information density
3. Comprehensive empirical evaluation on multiple datasets

**Weaknesses:**

1. Novelty issue:
- Core idea of wavelet-based tokenization for large models has been explored in previous work, e.g., Wave-ViT (Zhu ETL. 2024), though in different domains. Insufficient discussion of how this approach differs fundamentally from or improves upon prior works in wavelet-based tokenizer
2. Design justification:
- The marginal improvement over Chronos raises questions about whether the gains justify the additional benefits of wavelet-based tokenization. An ablation study comparing against simpler tokenization approaches would help validate the necessity of wavelets.
- Mismatch between extensive thresholding theory (Section 3.2) and empirical findings that no thresholding works best in the 8-gpu case.
3. Minor issues:
- The main text lacks a clear mathematical formulation of problem setup and wavelet decomposition, making it harder to understand the key transformation.
- Insufficient explanation of unexpected results in other normalization choices (Line 221)
- No clear shape information for approximation ($a_k$) and detail ($d_k$) coefficients (Line 225).
- Unclear explanation of length preservation in DWT (Line 227). It takes time to think if it is due to the downsampling.

[1] Zhu, Zhenhai, and Radu Soricut. "Wavelet-Based Image Tokenizer for Vision Transformers." arXiv preprint arXiv:2405.18616 (2024).

**Questions:**

1. How does your wavelet-based tokenization fundamentally differ from prior approaches like Wave-ViT, particularly in handling temporal vs spatial data? Are concepts like "pixel-space token embedding" fundamentally different from the wavelet-based tokens in TS?
2. The finding that single-level decomposition is optimal seems to contradict the motivation for using wavelets. Could you explain this apparent contradiction? Does this undermine the motivation for using wavelets?
3. Could you provide ablation studies comparing against simpler tokenization approaches to justify the added complexity of wavelets?
4. How sensitive is the model's performance to different batch sizes and computing configurations, given the significant difference between single-GPU and 8-GPU results?

---

> ### Author Response · Authors · 2024-11-21
> **OFFICIAL RESPONSE TO REVIEWER zVkm**
>
> Thank you for your comments and questions that helped improve our paper. We have summarized our main revisions and answers in response to all reviewers in a separate comment above. Below we provide point-by-point answers to each section in your review.
>
> ---
> Regarding
> > Core idea of wavelet-based tokenization for large models has been explored in previous work, e.g., Wave-ViT (Zhu ETL. 2024), though in different domains. Insufficient discussion of how this approach differs fundamentally from or improves upon prior works in wavelet-based tokenizer
>
> and
> > How does your wavelet-based tokenization fundamentally differ from prior approaches like Wave-ViT, particularly in handling temporal vs spatial data? Are concepts like "pixel-space token embedding" fundamentally different from the wavelet-based tokens in TS?
>
> Thank you for bringing this very recent paper to our attention. We have added a citation to this work in the Related Work section. Nonetheless, this work has been developed with a *completely different motivation* than ours and addresses a *different problem on another domain* (as you have pointed out),  i.e., image classification via Vision Transformers. As such, the conceptual similarity of WaveToken with the tokenizer introduced by Zhu. et al (2024) does not diminish the contribution or significance of our work. Furthermore, while conceptually similar, the image tokenizer in Zhu et al. (2024) is substantially different and cannot be directly compared to our work, which instead
> 1. proposes a tokenizer specifically tailored for time series forecasting with large foundation models based on the T5 architecture. The proposed design in Zhu et al. (2024) can’t be applied directly for T5 because 2D wavelet filters cannot be applied to time series;
> 2. leverages quantization of the wavelet coefficients in bins according to the coefficients’ distribution; and
> 3. preserves the length of the input and output sequences regardless of the decomposition level applied, thanks to the use of a maximally decimated wavelet transforms (see Appendix A.2).
>
> ---
> Regarding
> > The marginal improvement over Chronos raises questions about whether the gains   justify the additional benefits of wavelet-based tokenization. An ablation study comparing against simpler tokenization approaches would help validate the necessity of wavelets.
>
> and
> > Could you provide ablation studies comparing against simpler tokenization approaches to justify the added complexity of wavelets?
>
> The *core motivation of our work* was to develop a general purpose tokenizer that seamlessly captures global and local patterns in the time series. Our **qualitative results** (Figure 1 and Section 4.3) demonstrate the impressive performance of WaveToken on complex edge cases where existing foundation models fail, almost completely. These results highlight the remarkable ability of a wavelet-based tokenizer to capture nonstationary signals – which are pervasive in practical applications. Nevertheless, as outlined in the paper, **WaveToken also improves quantitatively** on Chronos 75% of the times and 83% of the times on the in-domain (15 datasets) and zero-shot (27 datasets) benchmarks, respectively, while employing a vocabulary one-quarter the size of Chronos. In addition, WaveToken achieves the *best average rank across all metrics for both Benchmarks I and II* (Figures 10 and 11).
>
> Note that our comparison with Chronos, which uses a simple uniform quantization scheme directly on the time series values, readily serves as an ablation against a simpler tokenization scheme (the simplest possible scheme in this context).
>
> We believe that in the natural progression of the field, different works make different types of contributions, and not every work brings (or needs to bring) a paradigm-shifting accuracy improvement. In the case of WaveToken, our findings position it as a promising avenue for developing general-purpose, information-efficient forecasting models, and it should therefore be evaluated independently in addition to its accuracy.

---

> > ### Author Response · Authors · 2024-11-21
> > **OFFICIAL RESPONSE TO REVIEWER zVkm (cont.)**
> >
> > Regarding
> > > Mismatch between extensive thresholding theory (Section 3.2) and empirical findings that no thresholding works best in the 8-gpu case.
> >
> > We believe there is no mismatch between Section 3.2 and empirical findings. Section 3.2 presents several thresholding methods commonly leveraged in the signal processing literature. These techniques, while useful, are usually developed for different downstream tasks such as compression or denoising, which are only tangential to time series forecasting. As explained in Section 4.4, lines 509-513, the empirical observation that thresholding does not improve performance is to be attributed to the quantization step of the tokenizer: all wavelet coefficients close to 0 are mapped to the bin centered at 0. This value is then used at inference time to forecast tokens mapped to this bin, thereby implicitly thresholding small coefficients. As such, there is no mismatch between our discussion of Section 3.2 and the empirical results. Nonetheless, we have clarified this point in the revision.
> >
> > ---
> > Regarding
> > > The finding that single-level decomposition is optimal seems to contradict the motivation for using wavelets. Could you explain this apparent contradiction? Does this undermine the motivation for using wavelets?
> >
> > The optimality of single-level decomposition in our context does not contradict the motivation for using wavelets, as the first-level wavelet coefficients already decompose the time series into high- and low-frequency components capturing global and local patterns.  In the signal processing literature, the number of levels for a wavelet decomposition is usually chosen empirically, and can also depend on the length of the input and output sequence. See also [1] for another application of a single-level wavelet decomposition to signal denoising.
> >
> > Going deeper into the decomposition provides finer granularity, but also imposes unnecessary structure in the concatenated coefficients used as input to the architecture. We indeed observed that the attention mechanism did not benefit from deeper decompositions, which may be a consequence of the autoregressive modeling setup which is forced to learn to attend to more groups of wavelet coefficients (see Section 4.4) making the modeling task more challenging.
> >
> > [1] Papakostas G. et al. (2015), “Two-Stage Evolutionary Quantification of In Vivo MRS Metabolites”
> >
> > ---
> > Regarding
> > > How sensitive is the model's performance to different batch sizes and computing configurations, given the significant difference between single-GPU and 8-GPU results?
> >
> > Training on 8 GPUs effectively means that the model sees 8 times more data, so the improved performance is a completely expected outcome. This is a standard observation, especially in the context of modern foundation models which are trained for only a few epochs (often even a single epoch), equivalent to an (almost) infinite data regime.

---

> ### Author Response · Authors · 2024-11-25
> **A Gentle Reminder**
>
> Dear Reviewer zVkm,
>
> Thank you again for the useful comments and questions that helped improve our paper. As the end of the discussion period is approaching, we would like to gently remind you to take into account our responses and the updated version of the manuscript into your evaluation.
>
> We hope that we have satisfactorily addressed your questions and concerns. If so, we request you to consider raising your score to reflect that. If you have further questions, we will be happy to respond to them.
>
> Thank you.
>
> Best Regards,
> The Authors

---

> ### Author Response · Authors · 2024-11-27
> **Follow-up Reminder**
>
> Dear Reviewer zVkm,
>
> Thank you again for your review. We kindly remind you to take into account our responses and the updated version of the manuscript into your evaluation.
>
> We hope that we have satisfactorily addressed your questions and concerns. If so, we request you to consider raising your score to reflect that. If you have further questions, we will be happy to respond to them.
>
> Thank you.
>
> Best Regards,\
> The Authors

---

> > ### Comment · Reviewer_zVkm · 2024-11-27
> >
> > Thank you for your response. I still have some concerns about isolating and validating the core contribution of the wavelet-based tokenization approach. The current evaluation does not convincingly isolate the benefits of the wavelet tokenizer. And the proposed model heavily builds on Chronos, inheriting much of its structure while introducing modifications on scaling and quantization methods. Therefore, it becomes difficult to attribute the performance gains solely to the wavelet tokenizer. While the authors claim the wavelet tokenizer is a general-purpose solution, the reliance on single-level decomposition and its entanglement with other key components (scaling and quantization) raises questions about its universal utility. Without fully leveraging the multi-resolution analysis traditionally associated with wavelets, the wavelet tokenizer’s contribution may not justify the added complexity, especially when compared to simpler alternatives (simpler means and differences).
> >
> > 1. The normalization choice remains unanswered
> >
> > The authors stated that the specific wavelet transform we adopt is not translation-invariant, hence a shift in the input signal can lead to different coefficients. However, I believe this statement may be somewhat misleading:
> >
> > - Scaling ensures consistent results across varying input scales but does not address or influence the translation-invariance of the wavelet transform itself. Translation-invariance refers to whether wavelet coefficients remain stable when the input signal is shifted **in time**, which is independent of scaling.
> > - The wavelet transform inherently operates on the shape of the input signal, not its absolute amplitude. While scaling (such as z-score) adjusts the signal amplitude, it does not change the underlying time-frequency structure of the wavelet coefficients.
> >
> > Furthermore, the authors claimed that other popular normalization techniques could therefore yield unexpected results for similar inputs. Can this statement be interpreted as evidence that other normalization choices were tested and found less effective than z-score scaling? If so, how significant were the performance differences? If z-score scaling provides optimal results, it should also be acknowledged as an important component of the model’s performance.
> >
> > 2. The justification for the benefits of the wavelet transform remains unconvincing
> >
> > - Using Chronos as an ablation experiment to justify the additional gains from wavelet-based tokenization is less fair, as the two models differ not only in the use of wavelets but also in their scaling and quantization methods:
> >     - Chronos employs mean scaling and a simple uniform quantization scheme.
> >     - The proposed model uses z-score scaling and the Freedman-Diaconis rule for quantization.
> > - These differences in vocabulary construction and preprocessing make it challenging to disentangle the specific contributions of the wavelet transform.
> >
> > Additionally, the authors claim that their quantization approach implicitly thresholds noise by mapping wavelet coefficients close to zero into a bin centered at zero. This property indicates that the quantization method plays a significant role in improving model performance by reducing the impact of noise. However, this adds another layer of complexity, as the observed gains may stem from the quantization method rather than the wavelet transform itself.
> >
> > The normalization choice also contributes to this complexity. Prior research, such as RevIN (Kim et al., 2021), has shown that z-score normalization is critical for time-series forecasting, particularly for datasets affected by distribution shifts. Given the importance of z-score normalization, it is reasonable to consider it as an independent factor contributing to the model’s performance. In contrast, Chronos relies on mean scaling, which is less effective in addressing distribution shifts. Consequently, the performance gains attributed to the wavelet tokenizer might instead result from the improved scaling method.
> >
> > 3. Concerns about the general-purpose utility of the wavelet tokenizer
> >
> > The reliance on single-level decomposition undermines the claim that the wavelet tokenizer is a general-purpose solution for time-series representation. By limiting the decomposition to a single level, this tokenizer reduces the wavelet transform’s functionality to a fixed filter size, restricting the model's ability to adaptively analyze global and local patterns, as it provides only one resolution to detect features. Moreover, the use of a single-level decomposition makes the tokenizer’s functionality easily replaceable by simpler approaches. For example, moving averages could effectively detect trends, differencing could identify localized changes, and direct filtering methods could extract fixed-resolution features—all without the added complexity of wavelet transforms.

---

> > > ### Author Response · Authors · 2024-11-28
> > > **Official Response to Remaining Concerns of Reviewer zVkm**
> > >
> > > Thank you for your response. We understand that *one of your primary concerns in the original review* was regarding the novelty of our work in the context of VIT models that use wavelets. We hope our initial reply has sufficiently addressed your concerns on this matter. If you have any further questions or require additional clarification, we would be more than happy to provide it. Otherwise, we would greatly appreciate it if you could kindly acknowledge our response regarding the novelty. Below, we address the follow-up points you raised.
> > >
> > > ---
> > > As to
> > > > 1. The normalization choice remains unanswered
> > >
> > > We apologize for the confusion in this matter, as the term translation-invariance can be ambivalently used for shifts in time and shifts in value. We agree that translation-invariance in **time** refers to coefficients being stable when the input signal is shifted in time domain, but in our work we referred to translation-invariance with respect to **values**. That is, when a signal $f(x)$ is shifted by a constant $c$, in which case the wavelet (approximation) coefficients also change. Z-score scaling makes the approximation coefficients – which capture low frequency content including the mean – insensitive to additive constants thanks to centering. In addition, z-score scaling makes sure that the time series has unit variance, thereby standardizing the magnitude of detail coefficients, which only capture relative variations. Overall, these effects ensure that the model is learning from representations of the inputs that are as consistent as possible for signals that effectively carry the same time-localized frequency content.
> > >
> > > We will update the manuscript accordingly to clarify these points.
> > >
> > > ---
> > > As to
> > > > 2. The justification for the benefits of the wavelet transform remains unconvincing
> > >
> > > Our work proposes a **novel tokenizer**, which by definition is a *combination of pre- and post-processing operations*. As such, normalization, wavelet transform and quantization should all be considered as part of a unique tokenization scheme, not separately. We argue and demonstrate that our combination of operations can be more effective to capture complex edge cases where existing foundation models fail almost completely, and to deliver an overall excellent performance on In-Domain and Zero-Shot benchmarks. This is the same terminology used in the Chronos paper.
> > >
> > > Regarding the **impact of the normalization choice**, we conducted an **additional experiment** to dispel any doubt. The table below compares the performance of Chronos (Small) using mean absolute scaling (MeanAbs) versus z-score scaling (ZScore). The latter clearly performs worse across all benchmarks and metrics, hence the ZScore scaling in itself cannot be deemed to be the source of the performance improvement we observe for WaveToken.
> > >
> > > | Benchmark  | Metric | Chronos (Small, MeanAbs) | Chronos (Small, ZScore) |
> > > |------------|--------|--------------------------|--------------------------|
> > > | In Domain  | WQL    | **0.596**            | 0.610                |
> > > | In Domain  | MASE   | 0.726            | **0.724**              |
> > > | In Domain  | VRSE    | **0.537**            | 0.571                |
> > > | Zero Shot  | WQL    | **0.661**            | 0.674                |
> > > | Zero Shot  | MASE   | **0.819**                | 0.820            |
> > > | Zero Shot  | VRSE    | **0.581**            | 0.596                |
> > >
> > > Regarding the **impact of the quantization scheme**, we note that the implicit thresholding effect is happening both for WaveToken and Chronos, which maps raw time series values to bin centers, thereby removing noise too. As such, even this mechanism cannot be deemed to be the source of the performance improvement we observe for WaveToken. Finally, we also note that the implicit removal of noise is not the central point of either method (Chronos or WaveToken), and in our case we only offered it as a potential justification for the empirical observation that various thresholding techniques were not found to be effective.

---

> > > > ### Author Response · Authors · 2024-11-28
> > > > **Official Response to Remaining Concerns of Reviewer zVkm (cont.)**
> > > >
> > > > As to
> > > > > 3. Concerns about the general-purpose utility of the wavelet tokenizer
> > > >
> > > > To clarify, we never claimed to provide a general representation method for time series (in the sense of representation learning), but only demonstrated that a wavelet-based tokenizer – embedded into a foundation model for time series – results in a robust, general-purpose **forecasting** model, which is demonstrated by our quantitative and qualitative results.
> > > >
> > > > Responding point-by-point:
> > > > - The **decomposition level** for a wavelet transform is *always* chosen empirically and the optimal level depends on the downstream task. In our case, the tokenizer is applied to forecast an *extremely wide variety of time series* in terms of frequencies, seasonalities, history and forecast lengths. As such, the tokenizer must strike an empirical balance between decomposing the high- and low-frequency structures in the inputs while providing a compact and easily learnable structure to the model architecture. Multiple resolution levels may be useful for some time series but can also be redundant for others. One could potentially tune the decomposition level individually if the model was trained on a single dataset, but on ~900K diverse time series – the cumulative size of our training set – a single scale can already explain much of the signal variance to produce accurate forecasts, which is the task we consider. Even a single decomposition level adds more information about the time series than just quantizing the values directly, as done in Chronos. Note also that increasing the decomposition level can potentially be harmful as it exacerbates boundary effects due convolution with wavelet filters of fixed size, which require non-trivial signal extension techniques [1].
> > > >
> > > > - Wavelets inherently capture both time and frequency information in a single step, and **specific wavelet bases** have been studied for decades to provide both compression and feature extraction functionalities, which *simpler techniques cannot achieve neither simultaneously nor as efficiently*. Trends, localized changes and sudden spikes, fixed-resolution features and more can all be detected and analyzed by wavelets via a unified framework that naturally adapts to the complexity of the underlying signal, instead of having to combine multiple tools. Note also that the discrete wavelet transform runs in linear time with respect to the input size, hence it does not add *computational complexity*.
> > > >
> > > > - Finally, our work only proposes **one possible tokenizer**. We understand and agree that there might be better solutions yet to be developed. With WaveToken, we aim at opening promising avenues and foster research in foundation models for time series towards this exciting direction.
> > > >
> > > > [1] Usevitch, Bryan E. "A tutorial on modern lossy wavelet image compression: foundations of JPEG 2000." IEEE signal processing magazine 18.5 (2001): 22-35.
> > > >
> > > > ---
> > > > Thank you again for your time and engagement on this. We hope our responses above have satisfactorily addressed your additional concerns. If so, we request you to consider raising your score to reflect that. If you have further questions, we would be happy to respond to them.

---

> > > ### Author Response · Authors · 2024-12-02
> > > **Final Reminder**
> > >
> > > Dear Reviewer zVkm,
> > >
> > > Thank you again for your review and follow-up questions. This is a gentle reminder that today is the final day for reviewers to provide comments on the forum. We hope our initial and follow-up responses have sufficiently addressed your concerns regarding our submission. Below, we summarize your main comments and our corresponding responses:
> > >
> > > - **Novelty of Our Work**: In your initial review, you expressed concerns about the novelty of our approach, noting that wavelets have been used in the context of vision tokenizers. However, this does not diminish the significance of WaveToken as it was developed with a distinct motivation and for a completely different domain—time series. We also highlighted several technical differences between WaveToken and Zhu et al. (2024). Please refer to our initial response for details.
> > > - **Normalization and Benefits of Wavelet Transform**: In your follow-up, you raised questions about the role of normalization and the benefits of using the wavelet transform. To address this, we conducted an additional ablation study using z-score normalization in Chronos. The results show that simply applying this normalization does not enhance Chronos' performance. This demonstrates that WaveToken's effectiveness stems from the entire tokenization pipeline—normalization, wavelet transform, and quantization—not just the normalization method.
> > >
> > > If our responses have addressed your concerns, we would greatly appreciate it if you could reconsider your initial score of 3. Thank you again for your time and feedback.

---

> ### Author Response · Authors · 2024-11-30
> **Reminder before End of Discussion Period**
>
> Dear Reviewer zVkm,
>
> Thank you again for your time and engagement. As you know, the discussion period will end in two days on Monday December 2nd. We would like to gently remind you to take into account our responses to your additional concerns.
>
> We hope our responses above have satisfactorily addressed your additional questions. If so, we request you to consider raising your score to reflect that. If you have further questions, we would be happy to respond to them.
>
> Thank you.

---

### Author Response · Authors · 2024-11-21
**Summary of Responses and Revisions**

We thank all the reviewers for their useful comments, suggestions and questions, which helped improve the quality of this work. Below we provide a summary of our answers and updates to the manuscript.

---
### **Concerns about “marginal improvements” over existing models**

Some reviewers raised concerns about “marginal improvements” in terms of forecasting metrics with respect to existing models, especially Chronos.

The *core motivation of our work* was to develop a general purpose tokenizer that seamlessly captures global and local patterns in the time series. Our **qualitative results** (Figure 1 and Section 4.3) demonstrate the impressive performance of WaveToken on complex edge cases where existing foundation models fail, almost completely. These results highlight the remarkable ability of a wavelet-based tokenizer to capture nonstationary signals – which are pervasive in practical applications. Nevertheless, as outlined in the paper, **WaveToken also improves quantitatively** on Chronos 75% of the times and 83% of the times – considering all model sizes and metrics – on the in-domain (15 datasets) and zero-shot (27 datasets) benchmarks, respectively, while employing a vocabulary one-quarter the size of Chronos. In addition, WaveToken achieves the **best average rank across all metrics** for both Benchmarks I and II (Figures 10 and 11).

We believe that in the natural progression of the field, different works make different types of contributions, and not every work brings (or needs to bring) a paradigm-shifting accuracy improvement. In the case of WaveToken, our findings position it as a promising avenue for developing general-purpose, information-efficient forecasting models, and it should therefore be evaluated independently in addition to its accuracy.

---
### **Performance on Long-Horizon Forecasting**
Reviewer KBzW asked to investigate the performance of WaveToken on long-horizon forecasting.

We conducted **additional experiments** (see the new Figure 12 and Appendix E in the paper) on a **long-horizon benchmark** constructed by increasing the forecast length $H$ of each dataset in Benchmark II (Zero-Shot, 27 datasets) by a factor of 2 and 3 (see Table 9 for original forecast lengths), so that the long horizon varies accordingly to the time series frequency. The datasets in Benchmark II without sufficient history in all time series to allow for longer horizons have been skipped, resulting in 27 and 20 datasets for the $H \times 2$ and $H \times 3$ settings, respectively. **WaveToken outperforms other foundation models across all three metrics** in the $H \times 2$ setting. When $H \times 3$, TimesFM performs better with respect to WQL only. All results for Chronos and WaveToken are averaged across three different seeds.

---

> ### Author Response · Authors · 2024-11-21
> **Summary of Responses and Revisions (cont.)**
>
> ### **Usage of cross-entropy loss when forecasting continuous data via next-token prediction**
>
> Some reviewers asked questions regarding the meaning, advantages and possible pitfalls of using the cross-entropy loss when forecasting continuous data.
>
> While unconventional, using the cross-entropy loss for continuous data is not a new idea and is known in the literature as **regression via classification** [1, 2]. Previous works have shown the benefits of this approach across tasks, such as audio modeling (Wavenet [3]), Deep RL [4], and recently in the context of forecasting (Chronos [5]). Although the cross-entropy loss removes topological information, the model implicitly learns the topology from the data through large scale training. Using a categorical distribution allows the model to be flexible to accommodate multimodalities, strong asymmetries, sudden shifts, and more, as the distribution is not constrained to a specific shape.
>
> Additionally, the use of cross-entropy loss allows to sidestep common issues with standard forecasting objectives (like MSE or quantile loss) such as their dependence on the scale of the inputs and on the presence of outliers, which can significantly degrade learning. Conversely, other approaches – e.g., DeepAR or more recent foundation models like Moirai – impose restrictive assumptions on the output forecasts via parametric distributions (usually Gaussian or Student’s-t), which are often not empirically valid. In our comparisons with these models, we already compare WaveToken against these alternative objective functions.
>
> [1] Torgo, L., & Gama, J. (1997). Regression using classification algorithms. Intelligent Data Analysis, 1(4), 275-292
>
> [2] Stewart, L., Bach, F., Berthet, Q., & Vert, J. P. (2023, April). Regression as classification: Influence of task formulation on neural network features. In International Conference on Artificial Intelligence and Statistics (pp. 11563-11582). PMLR
>
> [3] Van den Oord A., Dieleman S., Zen H., Simonyan K., Vinyals O., Graves A., Kalchbrenner N., Senior A., Kavukcuoglu K. (2016), WaveNet: A Generative Model for Raw Audio, arXiv preprint arXiv:1609.03499
>
> [4] Farebrother, J., Orbay, J., Vuong, Q., Taïga, A. A., Chebotar, Y., Xiao, T., ... & Agarwal, R. (2024). Stop regressing: Training value functions via classification for scalable deep rl. arXiv preprint arXiv:2403.03950
>
> [5] Ansari, A. F., Stella, L., Turkmen, C., Zhang, X., Mercado, P., Shen, H., ... & Wang, Y. (2024). Chronos: Learning the language of time series. arXiv preprint arXiv:2403.07815
>
> ---
> ### **Code Release**
> We plan to release a user-friendly research package complete with details on how to train and evaluate our tokenizer and models. Unfortunately, we could not share our code for review at this stage due to pending legal approvals.

---

### Meta-Review · Area_Chair_XhKp · 2024-12-21

**Metareview:**

The paper proposes a wavelet-based tokenized to facilitate the learning of "complex representations directly in the space of time-localized frequencies”. The method uses wavelet decomposition and trains an autoregressive model to predict the wavelet coefficients. The authors have shown that their method performs on par or better than deep learning models on a benchmark of 42 datasets.

During the discussion phase, issues were raised about the improvements brought by WaveToken, to which the authors responded by pointing to the qualitative results, and to improvements over Chronos in many of the datasets.
Reviewer zVkm remained unconvinced about the merits of the tokenizer, due to the method being built on Chronos itself and that some of the performance gains can be attributed to the choice of normalization. The authors did provide experiments showing the impact of the normalization choice showing that ZScore is actually worse than MeanAbs for Chronos. The experiments still show that normalization can affect performance, though presumably the authors have used the best normalization scheme possible for each method. My assessment is that there is still a lack of clarity as to where the performance gain comes from.

Reviewer qSp1 also expressed concerns about the actual performance improvements by the method, the lack of statistical significance (obtained through repeated experiments or bootstrapping), and certain over claims made by the authors with respect to their results.

While it is clear that no time series method can win on all benchmarks, it is not clear when exactly this method is expected to work well. Are there any settings/assumptions under which it is guaranteed (or at least more likely) that it will outperform contenders? Without this, it is just one of the many time series methods out there that works marginally better on a random set of the benchmarks.

I also agree with the reviewer’s ascertainment that, if one of the advantages of the method comes from using a compact vocabulary, then some advantage of learning this vocabulary should be presented (beyond just the alleged performance gain).

Overall, while this paper has some merits, I do not deem it ready for publication at this time.

**Additional Comments On Reviewer Discussion:**

The main points raised by the reviewers during the discussion period were the lack of clarity in terms of where performance gains come from with this method, the statistical significance of the results, and the benefits brought from the compact vocabulary.

Although the authors provided some additional experiments, these did not sufficiently alleviate the reviewers' concerns.

---

### Decision · Program_Chairs · 2025-01-22

Reject